

# Evaluation of dynamically downscaled near-surface mass and energy fluxes for three mountain glaciers, British Columbia, Canada

Mekdes Ayalew Tessema[1], Valentina Radić[1], Brian Menounos[2], and Noel Fitzpatrick[1]

[1]Earth Ocean and Atmospheric Sciences Department (EOAS), The University of British Columbia, Vancouver, Canada
[2]Natural Resources and Environmental Studies Institute and Geography Program, University of Northern British Columbia, Prince George, Canada

**Correspondence:** Valentina Radić (vradic@eoas.ubc.ca)

**Abstract.** Most models of glacier melt are forced by observed meteorological data in the vicinity of the glacier in question. In the absence of these observations, the forcing is commonly derived from statistical or dynamical downscaling of low resolution reanalysis datasets. Here we perform dynamical downscaling via the Weather Research and Forecasting (WRF) model in order to evaluate its performance against the observations from automatic weather stations (AWSs ) for three mountain glaciers in

the interior of British Columbia, Canada over several summer seasons. The WRF model, nested within the ERA-Interim global reanalysis, produced output fields at 7.5 km and 2.5 km spatial resolution for all glaciers, as well as 1 km resolution for one of the glaciers. We find that the surface energy balance (SEB) model, forced by WRF at 2.5 km, adequately simulates the AWS-derived seasonal melt rates despite large biases in the individual SEB components. Overestimation of the number of clear sky days in WRF at 2.5 km explains the positive bias in the seasonal incoming shortwave radiation. This positive bias, however,

is compensated by a negative bias in the seasonal incoming longwave radiation, and by an underestimation of sensible and latent heat fluxes. The underestimation of sensible heat fluxes down to -80 % of AWS-derived fluxes, as calculated by the bulk aerodynamic method, is due to the underestimated near-surface wind speeds. An increase of WRF spatial resolution from 7.5 to 1 km does not improve the simulation of downscaled variables other than near-surface air temperature. For relatively small glaciers (< 7 km length along the flowline), the grid spacing of $\geq$ 1 km is not fine enough to simulate the local cloud convection

and topographic wind effects (katabatics). Since the incoming radiative fluxes, as dominant drivers of seasonal melting, are relatively well simulated (within 10 % difference from observed fluxes) by ERA-Interim at 80 km spatial resolution, there is no need for further downscaling of these radiative fluxes. Temperature and precipitation downscaling remains an important step for the simulation of turbulent fluxes and surface albedo.

## 1   Introduction

Models of glacier melt typically require meteorological variables as input data. The traditional approach to obtain these data is through the deployment of an automatic weather station (AWS) at or in close vicinity to the glacier of interest. In the absence of measured meteorological variables at glacier surfaces, a common alternative approach to obtain the input data for melt models is to use global and/or regional reanalysis datasets (e.g., Radić et al., 2014; Huss and Hock, 2015). These reanalysis products are typically too coarse to represent atmospheric conditions for a small, alpine glacier, a problem commonly referred





to as 'scale mismatch' (Fowler and Wilby, 2007). In addition, the performance of reanalysis datasets is error prone over a complex terrain because the processes in the atmospheric boundary layer are too complex to be adequately parameterized and the observations needed for the data assimilation in the reanalysis models are often absent (e.g., Jarosch et al., 2010; Blacutt et al., 2015).

The scale mismatch and relatively poor performance of reanalysis products over complex terrain have triggered the development of correction schemes to be applied prior to the implementation of reanalysis data into glacier melt and mass balance models (e.g., Machguth et al., 2009; Jarosch et al., 2010; Kotlarski et al., 2010; Mölg and Kaser, 2011). These correction schemes, more commonly known as climate downscaling methods, can be split into two categories: (1) statistical downscaling - which relies on empirical (statistical) relationships between large scale predictors (e.g., output of a global climate model

(GCM) or reanalysis) and the local scale variables observed at a given site; and (2) dynamical downscaling - which applies a high-resolution regional climate model (RCM) nested within a GCM to directly simulate the local scale variables. Generally, the statistical downscaling is computationally inexpensive and, therefore, used to project long-term (e.g., centuries) transient runoff and mass balance changes in response to climate scenarios from GCMs (e.g., Radić and Hock, 2006; Matulla et al., 2009; Springer et al., 2013; Clarke et al., 2015). A critical disadvantage of statistical downscaling, however, is its assumption

that the empirically-derived relations between large- and local-scale meteorological variables will not change in the future (e.g., Zorita and Storch, 1999). Also, the method depends on the availability of local-scale observations which is often a serious limitation for glacierized complex terrain. Dynamical downscaling is physically-based; it explicitly models processes taking place in the atmosphere and at the surface-atmosphere interface. While observations are still needed for model evaluation, dynamical downscaling is not tuned by them as is the case for statistical downscaling. To use a RCM with sub-kilometer spatial resolution,

however, is computationally expensive which makes them unsuitable for long-term projections. The performance of dynamical downscaling is also sensitive to the choice of RCMs and their physics parameterizations (Mearns et al., 2014). Considering the pros and cons of statistical versus dynamical downscaling, an increasing number of studies make use of the 'coupled' or 'hybrid' approaches (e.g., Ekstroem et al., 2015; Laflamme et al., 2016; Yhang et al., 2017).

    A small number of studies explicitly evaluated the dynamical downscaling approach in simulating the surface energy balance

(SEB) at glacier surface. This approach was initially accompanied by statistical corrections or interpolation methods (e.g., Machguth et al., 2009; Paul and Kotlarski, 2010; Kotlarski et al., 2010), but the corrections became redundant with the use of RCMs that employ domains of varying spatial resolution (e.g., Mölg and Kaser, 2011; Mölg et al., 2012b; Claremar et al., 2012; Collier et al., 2013, 2015; Schanke et al., 2015; Wrzesien et al., 2015; Aas et al., 2016; Wu et al., 2016). A benchmark study (Mölg and Kaser, 2011) used the Weather Research and Forecast (WRF) model (Skamarock and Klemp, 2007) to force a

SEB model for a month in both the dry and wet season for a glacier on Mt. Kilimanjaro. The downscaled hourly variables at 0.8 km resolution showed strong correlation with AWS observations at the glacier. The successful model performance at the sub-kilometer resolution, however, was not corroborated at the coarser 3 km resolution. Another study applied the WRF model with a nesting scheme of 24 km, 8 km, and 2.7 km to simulate a 2-year surface mass balance for three glaciers in Svalbard (Claremar et al., 2012). Strong correlations with AWS data were obtained for most downscaled variables except the near-surface wind

speed. The increased horizontal resolution of the inner-most nest (2.7 km) did not improve the model performance relative



**Table 1.** Characteristics of the study glaciers and their automatic weather stations (AWSs).

| Glacier | Area (km$^2$) | Elevation range (m) | AWS coordinates | Observation period |
|---|---|---|---|---|
| Castle Creek | 9.5 | 1900 - 2800 | AWS: 53° 03' 03.0" N, 120° 26' 39.6" W | 01 Aug - 12 Aug 2010 |
| | | | AWS: 53° 03' 03.0" N, 120° 26' 39.6" W | 21 Aug - 16 Sep 2012 |
| Nordic | 5 | 2000 - 2900 | AWS: 51° 26' 03.6" N, 117° 41' 59.0" W | 12 Jul - 28 Aug 2014 |
| Conrad | 15 | 1800 - 3200 | AWS$_1$: 50° 49' 29.5" N, 116° 55' 20.9" W | 15 Jul - 05 Sep 2015 |
| | | | AWS$_2$: 50° 49' 23.0" N, 116° 55' 16.6" W | 16 Jul - 07 Sep 2015 |
| | | | AWS$_1$: 50° 49' 22.9" N, 116° 55' 11.7" W | 19 Jun - 28 Aug 2016 |
| | | | AWS$_2$: 50° 46' 55.9" N, 116° 54' 43.1" W | 16 Jun - 22 Aug 2016 |

to the 8 km resolution because the WRF model failed to resolve the key topographic effects (e.g., valley circulation, clouds, shading, and albedo). Increased vertical resolution did improve model simulations, probably due to more representative vertical profiles of wind speed and moisture transport from the open sea. Further advancement in the application of the WRF model in glacier studies came from Collier et al. (2013) and Collier et al. (2015) who developed a high resolution interactive model

of the glacier-atmosphere interface and applied it over the Karakoram region in the Northwestern Himalaya. The model was run for two summer months using three nested domains (33 km, 11 km and 2.2 km spatial grid). The modeled near-surface air temperature and wind speed accorded with AWS observations from a nearby glacier, but the incoming shortwave radiation was overestimated due to poorly simulated cloud cover, humidity and topographic shading. Precipitation was also poorly simulated, likely because the inner-most domain was too coarse to resolve important meteorological conditions (e.g. orographic uplift or

microscale complex flow features) that affect glacier mass balance.

   Our study seeks to further investigate the use of dynamical downscaling to simulate meteorological fields and energy fluxes at the mountain glacier surface. A primary objective is to evaluate how well WRF can replicate AWS-derived meteorological data that are required for SEB models. We focus on the comparison between observed and modeled meteorological variables and surface energy fluxes at a point-scale on three mountain glaciers in the interior mountains of British Columbia. Model

performance at this scale is also relevant for distributed or whole-glacier melt modeling which relies on the point-scale AWS observations being extrapolated across the glacier (e.g., Hock, 2005). In addition to the WRF evaluation, we examine the sensitivity of downscaled variables and modeled melt to (1) varying spatial resolution and landcover in the WRF model; and (2) the choice of albedo and surface roughness settings in the SEB model. We also compare the performance of the SEB model relative to a simple positive degree day (PDD) model when both models are forced by the downscaled fields. Finally, we briefly

investigate the potential for using the downscaled variables in distributed SEB modeling.





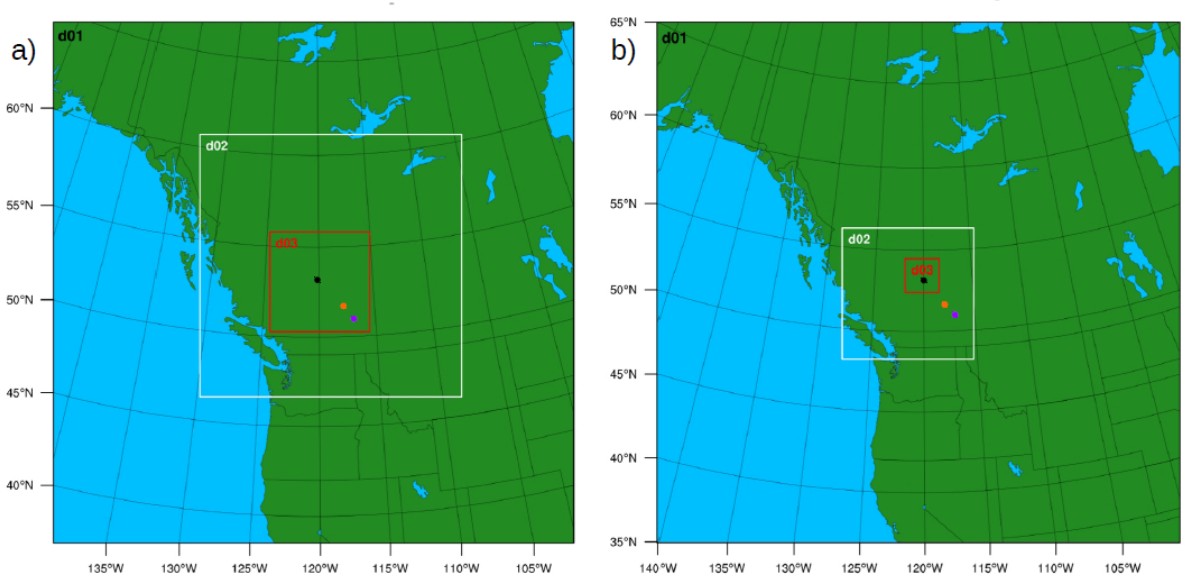

**Figure 1.** WRF domain setup for the study site with the three glaciers: Castle Creek (marked as black dot), Nordic (orange dot), and Conrad (purple dot). Nesting scheme from the outermost (d01) to the innermost domain (d03) with the following spatial resolution: (a) d01 = 22.5 km, d02 = 7.5 km and d03 = 2.5 km; (b) d01 = 25 km, d02 = 5.0 km and d03 = 1.0 km (for Castle Creek glacier only).

## 2 Data and Methods

### 2.1 Field Measurements

Our study area is situated in the interior mountains of British Columbia, Canada (Fig. 1), where the AWSs intermittently recorded data at three mountain glaciers within summer seasons of 2010-2016. Meteorological and glaciological measurements

from, in total, seven AWS sites at the three glaciers (Table 1) serve to evaluate the WRF model (Section 2.2) and to force the SEB model (Section 2.2). Castle Creek Glacier is located in the Cariboo Mountains, BC, and contributes meltwater to Castle Creek, a tributary of the Fraser River. Monitoring of the glacier's annual mass balance has been performed since 2009 (Beedle et al., 2015). In addition to the on-glacier AWS that operated during the ablation seasons of 2010 and 2012, two AWSs in the glacier vicinity have been in operation since 2007/2008 (Déry et al., 2010) measuring meteorological variables at two levels

(3 and 5 m above surface). The stations were situated near the glacier's transient snowline and in the glacier forefield. At the upper off-glacier AWS, the mean annual air temperature over 2007-2010 was -2.6 ° C while summer (June-August) mean air temperature was 6.6 ° C. Over the same period, total precipitation during summer was 94 mm at the upper off-glacier AWS, while mean monthly wind speeds often exceed 5 m s$^{-1}$ (Déry et al., 2010). The lower part of the glacier where the on-glacier AWS is located, is gently sloping with an approximate mean gradient of 7 °. Nordic and Conrad glaciers lie in the Purcell

Mountains in eastern BC and are located within the Columbia River basin. Their seasonal mass balance has been observed since 2012, but not yet published. The surface slope at the location of AWS on Nordic glacier is 13 °, while the location of



**Table 2.** Observed variables and their sensor specifications at Conrad glacier. Only variables used for the purpose of this study are listed.

| Variable | Sensor | Accuracy |
|---|---|---|
| Wind speed/direction | Young 05103ap Wind Monitor | $\pm$ 0.3 ms$^{-1}$ |
| Air temperature/humidity | Rotronic HC2 Probe and Shield (2015) | $\pm$ 0.1°C / 0.8 % |
| | Aspirated Rotronic HC2 Probe and Shield (2016) | $\pm$ 0.1°C / 0.8 % |
| Atmospheric Pressure | Vaisala PTB110 Barometer | $\pm$ 0.3 hPa |
| Precipitation | Texas Electronics Tipping Bucket Gauge | $\pm$ 1 % (up to 10 mm hr$^{-1}$) |
| Radiation fluxes | Kipp and Zonen CNR4 Radiometer | 10 - 20 Wm$^{-2}$ (pyranometer) |
| | | 5 - 15 Wm$^{-2}$ (pyrgeometer) |
| Surface height | CSI SR50A Sonic Ranger (three sensors per AWS) | $\pm$ 0.01 m |

AWSs on Conrad glacier have the slope of 8 ° in the ablation area and 3 ° in the accumulation area. Since the three glaciers lie within Canadian Rockies it is likely that they all share similar climatology. Surface area and elevation range of the three glaciers are listed in Table 1, as well as the location coordinates and operation dates of the AWSs. Observations from all three glaciers consist of incoming and outgoing components of shortwave and longwave radiation fluxes, air temperature and humidity, wind

speed and direction, turbulent fluxes, and surface height changes as an indicator of solid precipitation and ablation. At each site, the eddy-covariance method was used to derive estimates of roughness lengths for momentum, temperature and humidity. At Castle Creek glacier, the measured surface air pressure at the lower and upper off-glacier stations is interpolated to derive the pressure at the on-glacier AWS, while rainfall rate is taken from the lower off-glacier AWS. A melting ice surface was present during all the observations at Castle Creek glacier, with some intermittent fresh snowfall events. At Nordic glacier, a

transitional snow surface was present for the first four days, with partial snow cover diminishing to a fully bare ice surface. The details about the data (sensor type, sampling frequency, accuracy control) and atmospheric conditions at Castle Creek glacier are given in Radić et al. (2017), while the details from Nordic glacier are in Fitzpatrick et al. (2017).

At Conrad glacier, a total of four AWS deployments were executed during 2015 and 2016: two stations in the ablation zone from July to September 2015, and one in each of the ablation and accumulation zones from June to August 2016 (Table 1). A

melting ice surface was present during observations at both stations in 2015, and for most of the 2016 observation period at the AWS in the ablation zone. At the AWS in the accumulation zone a snow surface was present throughout the 2016 observation period, and for the first 10 days at the AWS in the ablation zone. The meteorological sensors were housed on a four-legged quadpod, which provided a stable platform (as monitored by inclinometer) that lowered as the ice melted, and maintained a nearly constant height of the sensors above the surface. Near-surface meteorological variables and fluxes were measured at a

nearly constant height above the surface (wind speed at 2.7 m, temperature and relative humidity at 2 m, and radiation variables at 1.6 m). All variables were saved as 1 min averages except for the rainfall which was saved as 1 min totals. Table 2 provides details on the sensor type and accuracy control at Conrad glacier. A time lapse camera in close proximity (10-30 m) to each AWS was used to observe the surface and atmospheric conditions over a season, and to monitor the station's behavior.



## 2.2 The Weather Research and Forecasting model

We ran the advanced research WRF model, version 3.8.1 (Powers et al., 2017), configured with three nested domains, with the inner-most domain covering the sites of all three study glaciers (Fig. 1). The parent domain (d01) of 22.5 km horizontal grid spacing covered the bulk of North America, while the nested domains (d02 and d03) had a horizontal grid spacing of 7.5 km

and 2.5 km, respectively (Fig. 1). Only the model output from the two nested domains was used for the analysis and henceforth referred to as $mod_{7.5}$ and $mod_{2.5}$ output, respectively. For Castle Creek Glacier we also ran the WRF model configured with three domains of 25, 5 and 1 km resolution. Only the model output on 1 km grid spacing was used (referred as $mod_{1.0}$) in order to test the sensitivity of results to an increase in spatial resolution from the 2.5 km to 1 km. All model runs used the same model specifications and input data (Table 3) following those from Mölg and Kaser (2011), Collier et al. (2013), and Collier

et al. (2015). As boundary conditions for the parent domains, we used ERA-Interim reanalysis (Dee et al., 2011) with 6-hr temporal resolution. ERA-Interim has been the most commonly used in the aforementioned studies and is shown to perform better than other reanalyses for Western Canada (Radić et al., 2015). Air temperature, wind speed and relative humidity values were taken from 37 pressure levels, in the range from 1 hPa (top level) to 1000 hPa (bottom level). Lower initial conditions include soil temperature and moisture at four levels, surface and sea-level pressure, and near-surface wind, temperature and

humidity. Sea surface temperature (SST) was set to a constant value, i.e., the time varying option was turned off. Similarly to other studies that used WRF on glacierized terrains, the one-way nesting approach was applied rather than the two-way nesting. In the one-way nesting, the information flows only from the outer (parent) domain to the nested domain, while there is no feedback from the nest domain solution to the parent. We employed the default lateral boundary control that included a four-grid-point relaxation zone with the outermost grid-point specified. Similarly to other studies we followed here, no nudging

or re-initialization was used within the parent and nested domains. All three model domains used the same vertical resolution (60 layers up to a model top of 50 hPa) and physical parameterizations (Table 3), with the exception of the cumulus convection scheme that was omitted in the inner-most domains (2.5 and 1 km). Due to inconsistencies between the physics in the RCMs and those given by the initial conditions, a model spin-up period was required to make sure that all RCM components reach a physical equilibrium (Montavez et al., 2017). We therefore applied a model spin-up of $\approx 3$ days for each model run, and the

WRF output was saved on a 3-hr time step covering the period of observations with AWSs (Table 1).

Figures 2 - 3 show the WRF model topography and landcover category data for each of the three glaciers in the nested domains of different spatial resolution. To compare the WRF output with the AWS data for each glacier, an output from the grid cell with the minimum horizontal distance to the AWS geographical coordinates was used. Despite the overlap in the location, the elevation of the selected grid cell differed from the actual AWS elevation (Table 4). Furthermore, none of the

selected grid cells that represent the AWS within the ablation area correctly captured the snow/ice land category (Fig. 2 - 3). Instead, the representative grid cells were classified either as evergreen needle-leaf forest or bare ground tundra. In addition to the selected AWS grid cell for each glacier, we used output from the grid cell with a snow/ice land category that is the closest, in horizontal and vertical, to the selected grid cell. The output from these grid cells, henceforth labeled as $mod*_{1.0}$ and $mod*_{2.5}$, was mainly used in the sensitivity tests of which, the details are explained in the sections below.





**Table 3.** WRF model configuration.

| Domain setup | | | |
|---|---|---|---|
| Horizontal grid spacing | 22.5, 7.5, 2.5 km (domains 1-3) | | |
| | 25, 5, 1 km (domains 1-3; Castle Creek glacier only) | | |
| Time step | 90, 30, 10s (domains 1-3) | | |
| | 180, 60, 20s (domains 1-3; Castle Creek glacier only) | | |
| Vertical levels | 60 | | |
| Model top pressure | 50 hPa | | |
| **Model physics** | | | |
| Radiation | RRTMG | Iacono et al. (2008) | |
| Cumulus | Grell 3D Ensemble | Grell (1993); Grell and Devenyi (2002) | |
| Microphysics | Thompson 2-moment | Thompson et al. (2008) | |
| Atmospheric Boundary Layer | Mellor-Yamada (MYNN) Level 3 | Nakanishi and Niino (2006) | |
| Surface layer | Mellor-Yamada (MYNN) Level 2.5 | Nakanishi and Niino (2009) | |
| Land surface | Noah-MP | Niu et al. (2011); Yang et al. (2011) | |
| **Dynamics** | | | |
| Top Boundary conditions | Upper level diffusive layer | | |
| Horizontal advection | 2nd order diffusion on model levels | | |
| **Lateral boundaries and input data** | | | |
| Specified boundary width | 5 | | |
| Forcing | ERA-Interim (T255 spectral resolution) | Dee et al. (2011) | |
| Land cover data | ESA (300 m) | ESA (2018) | |
| Topography | SRTM (90 m) | Jarvis et al. (2008) | |

**Table 4.** AWS elevation (units in meters above sea level): actual versus WRF grid cell for each study site.

| Site | Actual | $\text{mod}_{7.5}$ | $\text{mod}_{2.5}$ | $\text{mod}_{1.0}$ | $\text{mod}*_{1.0}$ | $\text{mod}*_{2.5}$ |
|---|---|---|---|---|---|---|
| Castle Creek 2010/2012 | 1967 | 2044 | 2185 | 1973 | 2371 | 2288 |
| Nordic 2014 | 2208 | 2021 | 2255 | - | - | 2325 |
| Conrad 2015 st1 | 2138 | 2250 | 2160 | - | - | 2488 |
| Conrad 2015 st2 | 2164 | 2250 | 2160 | - | - | 2488 |
| Conrad 2016 st1 | 2163 | 2245 | 2160 | - | - | 2488 |
| Conrad 2016 st2 | 2909 | 2194 | 2634 | - | - | 2488 |

## 2.3 Surface energy balance model

The selection of a SEB model followed from our goal to evaluate the WRF model by simulating the most relevant components of surface energy balance for a given grid point. To that end, we chose a SEB model of low complexity and forced it first with





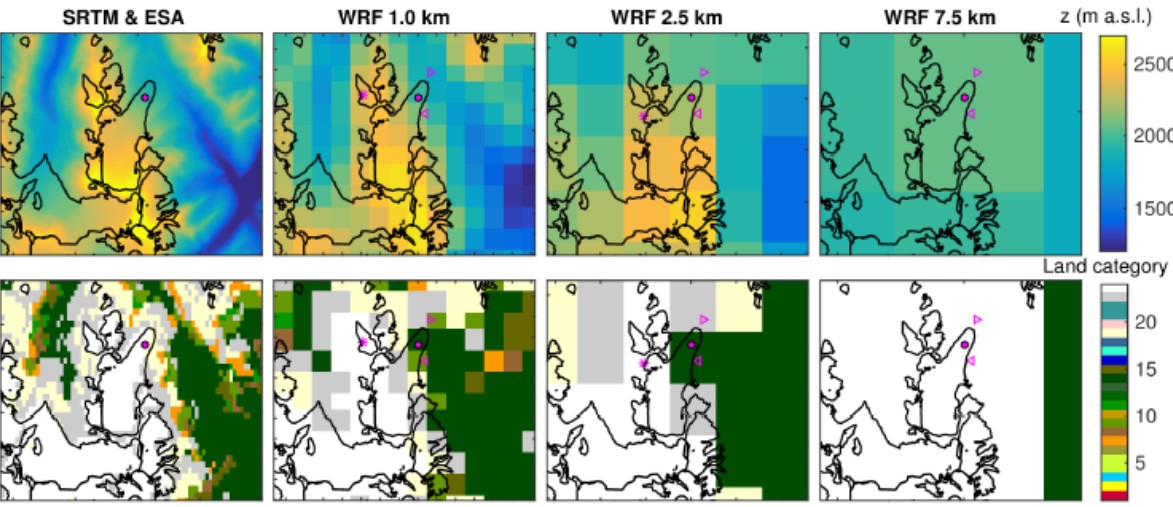

**Figure 2.** Topography and land category for the domain covering Castle Creek glacier from SRTM 30 m DEM and ESA CCI Landcover at 300 m resolution, respectively, in comparison to the topography and land category in the WRF inner domains at different spatial resolutions (7.5 km, 2.5 km, and 1 km). Categories from ESA CCI Landcover data are manually converted to the USGS 24-category Land Cover used in WRF (see http://edcdaac.usgs.gov/glcc/globe_int.html for the name list of land categories). Land category 24 (white color in the colorbar) is snow/ice. Markers indicate the on-glacier AWS (circle), two off-glacier AWSs (triangles), and the selected grid cell with ice/snow land category (star) to be used in the sensitivity test (see text). The outlines of glaciers (black lines) are from the Randolph Glacier Inventory.

the AWS data and then with the WRF output for the selected grid cells. In addition to 3-hourly melt rates, we also assessed the surface height changes as the difference between surface ablation and accumulation. The modeled surface height changes were compared to those measured by sonic rangers (SR50) at each study site.

    We used a modified version of the SEB model described in Fitzpatrick et al. (2017). The available energy flux for melting
5  ($Q_M$) at a given point was derived as:

$$Q_M = K_{in}(1-\alpha) + L_{in} + L_{out} + Q_H + Q_E, \tag{1}$$

in which the terms correspond to, from left to right: incoming shortwave radiation, broadband albedo, incoming and outgoing longwave radiation, and turbulent fluxes of sensible and latent heat. By convention, the fluxes into (out of) the glacier surface are considered positive (negative). $Q_E$ refers to latent heat associated with water vapor while the latent heat associated with
10  the melting and refreezing is part of $Q_M$. We neglected the ground heat flux and heat flux from precipitation, since both gave negligible contributions to the total melt over a melt season on Nordic Glacier (Fitzpatrick et al., 2017). Assuming a melting glacier surface ($T_0$=0°C), the longwave outgoing radiation, $L_{out}$, was approximated using the Stefan-Boltzmann law with emissivity set to unity. The turbulent heat fluxes were parameterized via the commonly used bulk aerodynamic method (e.g.,





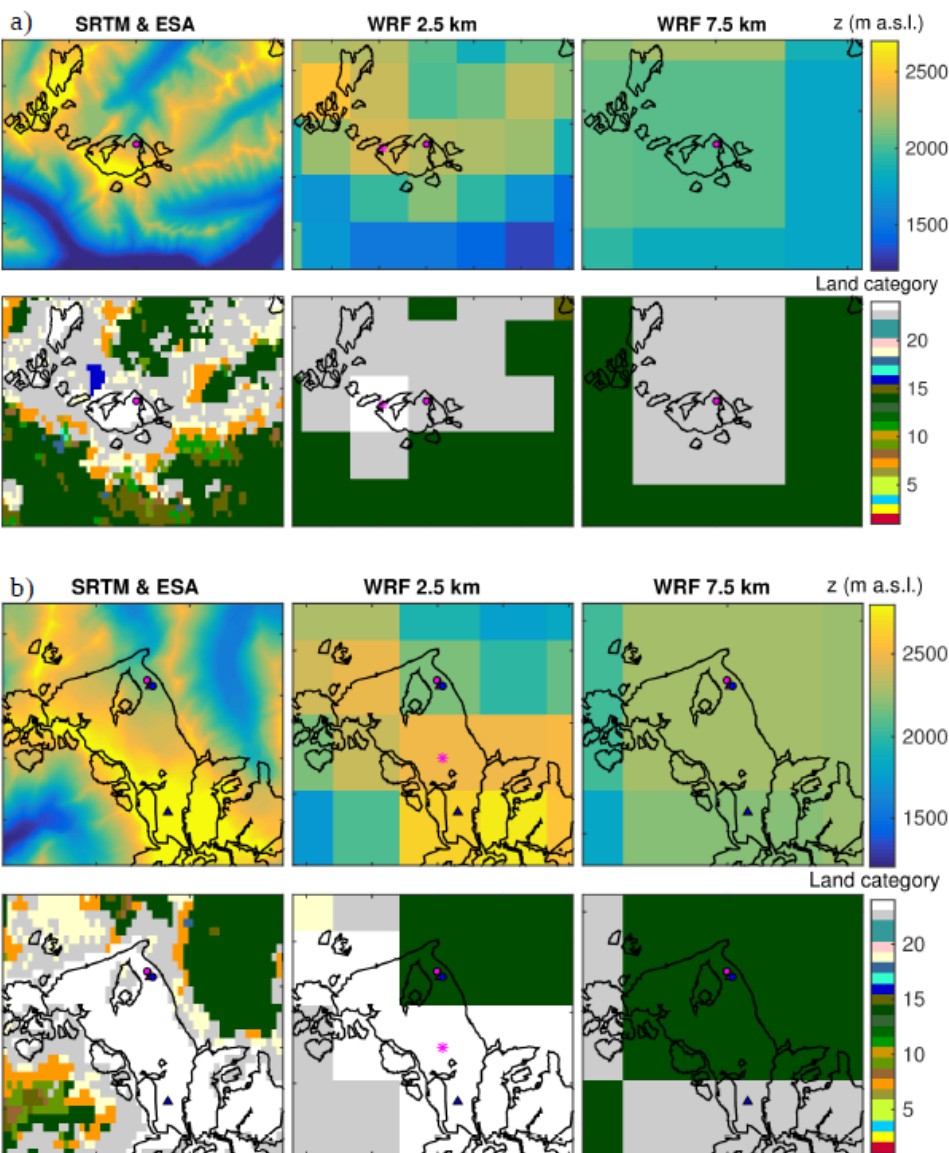

**Figure 3.** Topography and land category for the domain covering (a) Nordic glacier and (b) Conrad glacier from SRTM 30 m DEM and ESA CCI Landcover at 300 m resolution, respectively, in comparison to the topography and land category in the inner domains at different spatial resolutions (7.5 km and 2.5 km). Markers in indicate the on-glacier AWS (circle) and the selected grid cell with ice/snow land category (star) to be used in the sensitivity test. Markers in (b) indicate the two on-glacier AWSs in 2015 (pink circle and pink triangle) and in 2016 (blue circle and blue triangle), and the selected grid cell with ice/snow land category (star).

Conway and Cullen, 2013; Fitzpatrick et al., 2017; Radić et al., 2017):

$$Q_H = \frac{p}{p_0}\, \rho_a\, c_p\, C_H\, U_z\, (T_z - T_0), \tag{2}$$

$$Q_E = \frac{0.622}{p_0}\, \rho_a\, L_v\, C_E\, U_z\, (e_z - e_0), \tag{3}$$



where $U_z$ is mean (i.e. time-averaged) near-surface wind speed at height $z$, $T_z$ and $e_z$ are mean air temperature and vapor pressure at height $z$, respectively, while $T_0$ and $e_0$ are mean surface temperature and vapor pressure, respectively. $\rho_a$ is air density, $c_p$ is specific heat capacity of air at constant pressure (J kg$^{-1}$ K$^{-1}$), $L_v$ is the latent heat of vaporization of snow or ice (J kg$^{-1}$ K$^{-1}$), $p_0$ is the air pressure at standard sea level (1013 hPa), $p$ is the actual air pressure (hPa), and $C_H$ ($C_E$) is the

dimensionless exchange coefficient for sensible (latent) heat. $e_z$ is calculated from the relative humidity using the Clausius-Clapeyron equation. We parametrize $C_H$ ($C_E$) following the Monin-Obukhov stability theory, where the exchange coefficients depend on the surface roughness for momentum ($z_{0v}$), temperature ($z_{0T}$) and humidity ($z_{0q}$), and on the stability conditions in the surface boundary layer. Details about the parameterizations used in the SEB model can be found in Fitzpatrick et al. (2017).

The albedo in Eq. (1) was expressed as the ratio of observed daily totals (local daylight hours) of reflected and incoming shortwave radiation ($K_{out}/K_{in}$). The roughness lengths ($z_{0v}$, $z_{0T}$, $z_{0q}$) at each AWS site were estimated from the eddy-covariance method (see Radić et al. (2017) and Fitzpatrick et al. (2017) for details) and assumed constant over the observation period (Table 5). These were our default settings for albedo and roughness length values in the SEB model forced either by the AWS data or the WRF model output. In addition, we introduced an alternative setting for albedo where the 3-hr mean albedo

for ice, old snow and fresh snow was prescribed to 0.3, 0.65 and 0.8, respectively (Fig. 4). These values reflect previously used values well (Mölg et al., 2012b; Collier et al., 2013, 2015). In this scheme, the albedo of fresh snow (0.8) was prescribed only during active snowfall in the observation period. An alternative setting for surface roughness was: $z_{0v} = 10^{-3}$ m for the momentum surface roughness, while the scalar roughness lengths were determined from the surface renewal model of Andreas (1987). Combinations of the two settings for albedo and the two settings for roughness lengths yielded four different choices of

parameter settings to run the SEB model and derive $Q_M$: (1) observed albedo and observed roughness, (2) prescribed albedo and observed roughness, (3) observed albedo and prescribed roughness, and (4) prescribed albedo and prescribed roughness.

For a glacier's surface temperature at 0°C, a positive $Q_M$ in Eq. (1) drives melt. The total melt $M$ (in units of m water equivalent) integrated over a given time period was expressed as:

$$M = \begin{cases} \dfrac{Q_M}{\rho_w L_f}, & Q_M > 0 \\ 0, & Q_M \leq 0, \end{cases} \qquad (4)$$

where $\rho_w$ is water density, and $L_f$ (J kg$^{-1}$ K$^{-1}$) is the latent heat of melting or fusion. We converted the modeled melt into surface lowering, $z_M$, as:

$$z_M = M \frac{\rho_{f/i}}{\rho_w}, \qquad (5)$$

where $\rho_{f/i}$ is ice density ($\rho_i$=900 kg m$^{-3}$; Cuffey and Paterson, 2010) or the density of old (winter) snow, where the latter only applied at the AWS in the accumulation zone of Conrad glacier. In the absence of measured snow density in the accumulation

area of Conrad glacier we tuned its value by minimizing the root-mean-square-error (RMSE) between the 'measured' (via SR50) and modeled $z_M$ using a range of $\rho_f$ from 600 to 900 kg m$^{-3}$ with 50 kg m$^{-3}$ increment. The tuning resulted in $\rho_f$=800 kg m$^{-3}$. We note that (1) this value does not necessarily represent the actual snow density value as would be measured



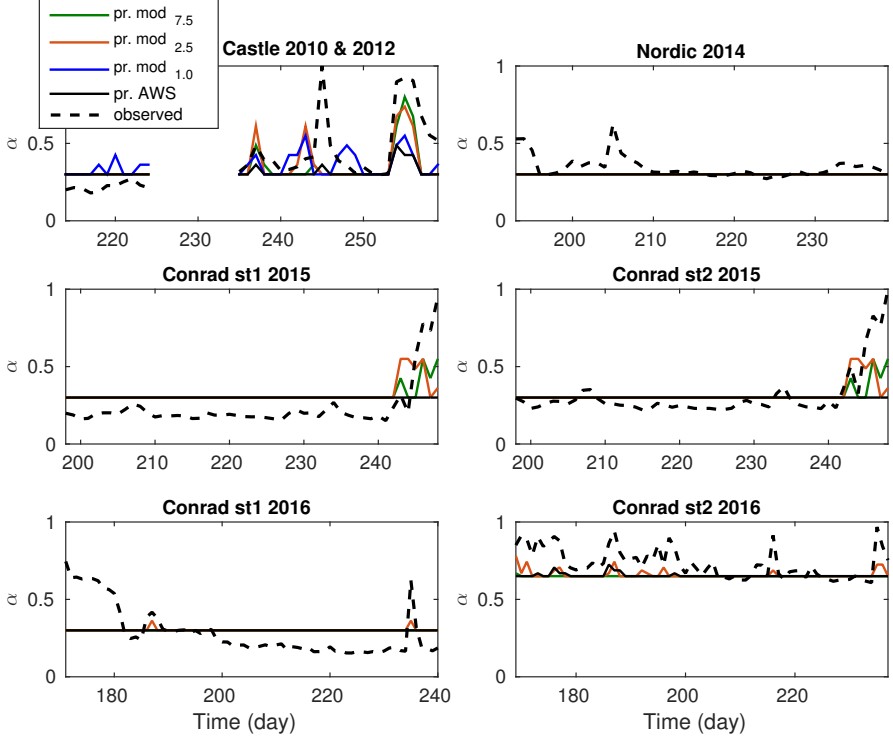

**Figure 4.** Mean daily albedo as observed from AWS (dashed) and as calculated (solid line) from the 3-hr albedo values prescribed for ice (0.3), old snow (0.65), and fresh snow (0.8). Note that the prescribed albedo values differ among the AWS and model runs due to differences in observed or modeled snowfall periods.

at the site, and (2) the 'measured' $z_M$ is approximated from the measured net surface changes that, in addition to melting, reflects accumulation and sublimation. The choice of $\rho_f$ has no impact on the evaluation results since the same value was used in the SEB model forced by AWS and WRF.

5   To directly compare the net surface height change ($z_{net}$) with SR50 data, in addition to the modeled melt, we took into account the modeled sublimation and fresh snow accumulation. Prior to the comparison, we applied filtering on the SR50 raw data to remove the high-frequency noise (Fitzpatrick et al., 2017). For the sites with more than one SR50 sensor (Conrad glacier sites), we inter-compared the filtered data among the sensors and manually removed any erroneous data associated with the infrastructure titling (usually occurring towards the end of the observational period). As the final SR50 data, we used the mean filtered signal across the sensors at the site. The sublimation ($z_S$; m w.e.) was calculated only when the latent heat flux ($Q_E$)

10   was negative:

$$z_S = |Q_E| \frac{\rho_{f/i}}{\rho_w \, L_s},\tag{6}$$



**Table 5.** Roughness length for momentum ($z_{0v}$; m), temperature ($z_{0T}$; m) and humidity ($z_{0q}$; m) used for each site, densities of fresh snow (in kg m$^{-3}$) derived from the tuning of modeled versus measured surface lowering (see text), and calibrated site-specific melt factor ($f_m$; mm w.e. $^\circ$C$^{-1}$ day$^{-1}$) in the positive degree-day model.

| Glacier and year | $z_{0v}$ | $z_{0t}$ | $z_{0q}$ | $\rho_s$ | $f_m$ |
|---|---|---|---|---|---|
| Castle Creek 2010 | $10^{-2.5}$ | $10^{-4.5}$ | $10^{-4.0}$ | 100 | 7.42 |
| Castle Creek 2012 | $10^{-2.5}$ | $10^{-4.5}$ | $10^{-4.0}$ | 200 | 5.99 |
| Nordic 2014 | $10^{-2.5}$ | $10^{-5.0}$ | $10^{-6.0}$ | 200 | 5.29 |
| Conrad 2015 AWS$_1$ | $10^{-2.5}$ | $10^{-4.0}$ | $10^{-3.5}$ | 200 | 6.80 |
| Conrad 2015 AWS$_2$ | $10^{-3.0}$ | $10^{-4.5}$ | $10^{-3.5}$ | 200 | 5.69 |
| Conrad 2016 AWS$_1$ | $10^{-3.0}$ | $10^{-4.5}$ | $10^{-4.5}$ | 200 | 7.63 |
| Conrad 2016 AWS$_2$ | $10^{-2.5}$ | $10^{-5.0}$ | $10^{-5.0}$ | 50 | 4.09 |

where $L_s$ (J kg$^{-1}$ K$^{-1}$) is latent heat of sublimation. Finally, we introduced a simple accumulation model to account for any surface height increase due to a fresh snowfall, $z_A$ (m w.e.), using the precipitation totals ($P$; m):

$$z_A = \begin{cases} P\dfrac{\rho_s}{\rho_w}, & T_z \leq \Phi \\ 0, & T_z > \Phi, \end{cases} \tag{7}$$

where $\rho_s$ is a fresh-snow density and $\Phi$, a threshold temperature for differentiating between liquid or solid precipitation, was

set to 0$^\circ$C. The net surface height change ($z_{net}$; m w.e.) was then derived as $z_{net} = z_A - z_M - z_S$, where negative (positive) $z_{net}$ stands for net surface lowering (rising). Similarly to the determination of $\rho_f$, the fresh-snow density ($\rho_s$) was tuned by minimizing RMSE between measured and modeled $z_{net}$ using a range of $\rho_s$ from 10 to 250 kg m$^{-3}$ with a 10 kg m$^{-3}$ increment. Final values for the snow densities at each AWS are listed in Table 5. Note that the model can only give a crude estimate of surface height changes since the mass balance model is oversimplified and we lack any actual measurements of

fresh-snow densities. Because of these simplifications, we also did not analyze the sensitivity of results to the choice of $\Phi$. The comparison of modeled versus measured surface height changes, nevertheless, served as a bulk quality control for the SEB model at each AWS site.

## 2.4 Model evaluation and sensitivity tests

Our first step in the analysis of WRF performance was to compare the timeseries of downscaled variables with the observed

variables at each site. The model evaluation was performed with the use of standard evaluation metrics: RMSE, mean-bias-error (MBE) and Pearson correlation coefficient (r). These metrics were quantified for the 3-hourly and daily timeseries of the following variables: 2m air temperature ($T_{2m}$), relative humidity ($RH_{2m}$), wind speed ($U$), surface air pressure ($SP$), precipitation ($P$), incoming shortwave radiation flux ($K_{in}$), incoming longwave radiation flux ($L_{in}$), sensible ($Q_H$) and latent heat ($Q_E$) fluxes, as assessed by the bulk method, and melt energy flux ($Q_M$), as derived from the SEB model (Eq. (1)). Note

that the original near-surface output from WRF, i.e. without any post-processing or corrections, gave wind speeds at 10 m



height above the surface while the AWS wind measurements were from $\approx 2$ m height above the surface. While one could apply a wind-speed correction for the height difference based on the logarithmic-wind profile assumption (e.g., Claremar et al., 2012), we chose not to since our goal was to evaluate the model skill without the use of any data correction. Specifically for the wind speed, the use of corrections that assume the logarithmic-wind profile would likely misrepresent the actual wind

profile during the prevailing katabatics, with low-level wind speed maxima, which are commonly observed at our sites (e.g., Fitzpatrick et al., 2017; Radić et al., 2017).

In addition to the evaluation of WRF against the AWS data, we analyzed the downscaling performance relative to the reference data that was being downscaled, i.e the ERA-Interim reanalysis. To that end, all the variables needed for the forcing of the SEB model were taken from the ERA-Interim output at $0.75° \times 0.75°$ spatial resolution, i.e. the dataset representing the

boundary and initial conditions to the WRF model. Following Wilks (2006) we calculated a skill score ($SSC$):

$$SSC = 1 - \frac{\text{MSE}}{\text{MSE}_{\text{ref}}}, \qquad (8)$$

where MSE is the mean squared error between AWS-derived and WRF-derived variables, while $\text{MSE}_{\text{ref}}$ is the mean squared error between AWS-derived and ERA-Interim variables. If the $SSC > 0$ for a given variable, the model performs better than the reference (ERA-Interim) model. Inter-comparison of $SSC$ for different spatial resolution in WRF explained whether the

finer horizontal spacing adds skill to the simulation, i.e. we tested the sensitivity of the downscaled variables to the choice of spatial resolution in WRF.

Another sensitivity test consisted of analyzing the impact of landcover in WRF on the downscaled variables. Since the sensitivity to landcover was not the main focus of this study, the test was relatively narrowly defined. We looked into the WRF output from the closest snow/ice grid cell (labeled as $\text{mod}*_{2.5}$ and $\text{mod}*_{1.0}$) to the grid cell representing the AWS at each site,

which was classified as forest or bare tundra (Figures 2 to 3). Comparing this output with the original WRF output, however, only partially addressed the effect of landcover on model performance since $\text{mod}*_{2.5}$ and $\text{mod}*_{1.0}$ output was also affected by grid-cell elevation. The comparison of model output with AWS measurements also depends on how close the selected grid cell is to the AWS location. To isolate the effect of landcover, we manually altered the land category of the AWS grid cell to snow/ice category in the model's inner-most domain and then reran the WRF model. In this way we could directly compare the

output for the same grid cell but with a different land category. This sensitivity test was only performed for Nordic Glacier using the run with the inner-most nest of 2.5 km grid spacing, and the modified AWS grid cell output was labeled as $\text{mod}_{2.5-\text{modified}}$.

The next step was to analyze the downscaling performance in the melt model over the observation period for each site. We compared the cumulative melt ($z_M$), as well as the cumulative net surface changes ($z_{net}$), derived when the SEB model was forced by the AWS measurements and then by the WRF output. We investigate the energy partitioning, where the daily

melt energy flux ($Q_M$), averaged over the observational period, was partitioned into the main SEB components: net shortwave radiation flux, net longwave radiation flux, and sensible and latent heat fluxes. We then tested the sensitivity of the results to the SEB model setup, using the four settings for albedo and roughness lengths, i.e. observed versus modeled (prescribed) values.

Finally, we compared the performance of the SEB model with a simple positive degree-day (PDD) model when the downscaled variables force both models. This analysis was motivated by the fact that current models of glacier mass balance on





regional and global scales use temperature-index models (e.g., Marzeion et al., 2012; Radić et al., 2014; Huss and Hock, 2015) or semi-empirical SEB models (e.g. Giesen and Oerlemans, 2012), all requiring fewer input variables than the SEB model used in our study. On the other hand, these empirical models require glacier-specific parameters (e.g., melt factors) or calibration with available observations of glacier mass balance. The melt factors are most commonly calibrated using observations

of seasonal mass balance or surface height changes (Hock, 2003). While we do have SR50 measurements of surface height changes at our sites, these measurements are post-processed and smoothed and, as a result, do not consistently resolve daily fluctuations in the surface lowering throughout the observation period. Furthermore, in order to represent only the measurements of surface lowering due to glacier surface melting, the SR50 measurements would need to be corrected for the episodes of accumulation and melting of the fresh snow. As is assumed in the PDD model, the melting process only reflects melting of

ice or old (last winter) snow and ignores the fresh fallen snow during ablation season. For all these reasons, rather than using the post-processed SR50 data for the calibration of the PDD model, we used the daily melt rates derived from the SEB model when forced with the AWS observations. By linearly regressing the daily SEB melt rates against the daily sum of positive 3-hr observed temperatures, we derived a melt factor value for each site and observation period (Table 5). These melt factors were then used in the PDD model forced with the WRF temperatures to derive the modeled cumulative melt over the observation

period for each site.

## 3 Results and discussion

### 3.1 Simulated versus observed atmospheric conditions and melt rates

We first present the results of the inter-comparison between AWS-derived timeseries and those from WRF and ERA-Interim. For the sake of brevity, in Figures 5 - 8 we show only the key results derived from the four AWS sites: Castle 2014, Nordic

2014, Conrad 2015 station 1, and Conrad 2016 station 2 (accumulation area). Table 6 shows the results of the inter-comparison for daily timeseries (RMSE, MBE and r) between $mod_{2.5}$ and AWS for all the sites. The following results apply to all seven sites:

**Air temperature and relative humidity:** Both ERA-Interim and the WRF model capture well the observed inter-diurnal variability in 2-m air temperature ($T_{2m}$). Amplitude of the diurnal cycle in temperature is, however, more pronounced

in the model runs than in the observations (Fig. 5) because the grid cell representing the AWS has a different category from snow/ice. The local cooling effect due to the presence of snow/ice at the surface is, therefore, not captured in the modeled timeseries. In the sensitivity tests, when the neighboring snow/ice covered grid cell is used ($mod_{2.5}^*$, $mod_{1.0}^*$) instead of the AWS grid cell, the diurnal cycle of temperature resembles more closely the observed one (not shown). Half the sites display a warm temperature bias in $mod_{2.5}$ (maximum MBE of 1.6 K across the sites; Table 6). Modeled daily

timeseries of 2-m air temperature are strongly correlated with observations (r $>$ 0.8 $mod_{2.5}$ across the sites; Table 6). Timeseries of relative humidity ($RH_{2m}$), due to its dependence on temperature, shows similar comparison of modeled



**Table 6.** Evaluation results for $mod_{2.5}$ against the AWS variables: r, RMSE and MBE derived from daily timeseries. Bold r values indicate values significantly different from zero at 0.05 significance level.

| | | Castle Creek | | Nordic | Conrad 2015 | | Conrad 2016 | |
|---|---|---|---|---|---|---|---|---|
| | | 2010 | 2012 | 2014 | st1 | st2 | st1 | st2 |
| **Variables** | **Units** | | | **r** | | | | |
| $T_{2m}$ | | **0.92** | **0.77** | **0.94** | **0.91** | **0.90** | **0.81** | **0.92** |
| $RH_{2m}$ | | **0.89** | 0.03 | **0.69** | **0.62** | **0.63** | **0.70** | **0.76** |
| $U$ | | -0.38 | **0.56** | -0.06 | 0.10 | 0.05 | -0.21 | **0.59** |
| $SP$ | | **0.95** | **0.92** | **0.98** | **0.98** | **0.98** | **0.94** | **0.95** |
| $K_{in}$ | | **0.60** | 0.38 | **0.68** | **0.63** | **0.61** | **0.46** | **0.32** |
| $L_{in}$ | | 0.49 | 0.37 | **0.50** | **0.57** | **0.53** | **0.27** | **0.36** |
| $Q_H$ | | -0.12 | **0.77** | -0.02 | **0.31** | **0.31** | -0.05 | **0.76** |
| $Q_E$ | | -0.01 | **0.45** | **0.48** | **0.51** | **0.51** | **0.26** | **0.73** |
| $Q_M$ | | **0.61** | **0.81** | **0.66** | **0.75** | **0.76** | **0.73** | **0.89** |
| | | | | **RMSE** | | | | |
| $T_{2m}$ | ° C | 1.8 | 2.5 | 1.8 | 1.5 | 1.6 | 2.4 | 1.3 |
| $RH_{2m}$ | % | 10.7 | 19.6 | 12.5 | 13.9 | 15.3 | 6.5 | 8.3 |
| $U$ | m s$^{-1}$ | 2.6 | 1.5 | 1.9 | 1.7 | 1.9 | 2.2 | 2.1 |
| $SP$ | hPa | 19.2 | 21.4 | 4.9 | 2.3 | 1.4 | 1.6 | 22.8 |
| $K_{in}$ | W m$^{-2}$ | 80.7 | 77.6 | 75.3 | 93.4 | 95.3 | 88.2 | 94.6 |
| $L_{in}$ | W m$^{-2}$ | 31.4 | 38.9 | 27.5 | 31.5 | 28.2 | 29.8 | 27.7 |
| $Q_H$ | W m$^{-2}$ | 49.0 | 32.7 | 49.7 | 44.8 | 36.7 | 34.4 | 18.9 |
| $Q_E$ | W m$^{-2}$ | 15.8 | 14.6 | 11.2 | 12.9 | 10.8 | 9.7 | 9.4 |
| $Q_M$ | W m$^{-2}$ | 81.0 | 45.5 | 56.1 | 59.4 | 50.8 | 47.5 | 17.6 |
| | | | | **MBE** | | | | |
| $T_{2m}$ | ° C | -0.1 | -0.8 | 1.4 | -0.2 | -0.4 | 1.6 | 0.4 |
| $RH_{2m}$ | % | 9.4 | 4.2 | 5.8 | 9.4 | 11.7 | -0.6 | 2.3 |
| $U$ | m s$^{-1}$ | -2.3 | -1.1 | -1.3 | -1.2 | -1.4 | -1.8 | -1.9 |
| $SP$ | hPa | -19.1 | -21.3 | -4.8 | -2.1 | -1.0 | 0.7 | 22.8 |
| $K_{in}$ | W m$^{-2}$ | 13.7 | 20.4 | 37.9 | 65.8 | 67.3 | 53.8 | 38.8 |
| $L_{in}$ | W m$^{-2}$ | -24.5 | -28.8 | -18.9 | -26.2 | -20.6 | -24.1 | -9.9 |
| $Q_H$ | W m$^{-2}$ | -42.3 | -23.9 | -35.4 | -31.9 | -27.1 | -26.4 | -12.4 |
| $Q_E$ | W m$^{-2}$ | -12.2 | -4.4 | -1.2 | 1.0 | 2.6 | -5.5 | 0.7 |
| $Q_M$ | W m$^{-2}$ | -55.3 | -22.0 | -22.4 | 1.6 | 9.2 | -8.0 | -7.5 |





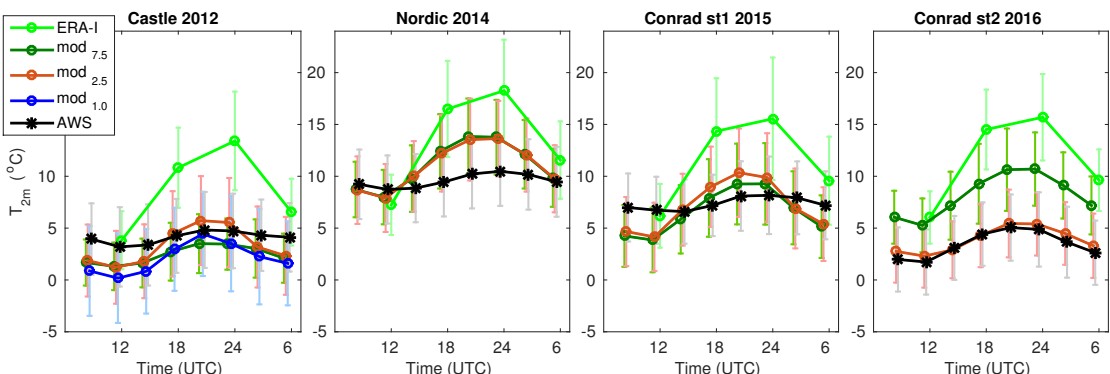

**Figure 5.** Mean daily cycle of 2m air temperature averaged over the observation period for each site and for: ERA-Interim (ERA-I), WRF at 7.5 km (mod$_{7.5}$), WRF at 2.5 km (mod$_{2.5}$), WRF at 1 km (mod$_{1.0}$), and AWS. Error bars represent the $\pm$ one standard deviation for the given hour of the day. AWS and WRF values are on 3-hr time step, while ERA-Interim values are on 6-hr time step.

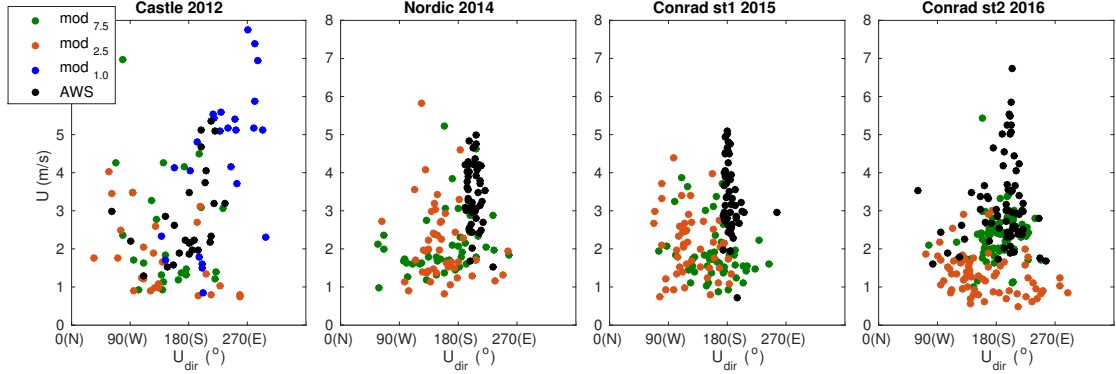

**Figure 6.** Mean daily wind speed ($U$) versus wind direction ($U_{dir}$) for each site and for: WRF at 7.5 km (mod$_{7.5}$), WRF at 2.5 km (mod$_{2.5}$), WRF at 1 km (mod$_{1.0}$), and AWS.

versus observed values as the $T_{2m}$ timeseries. A statistically significant correlation in daily $RH_{2m}$ (r > 0.6; p-value < 0.05) is achieved for all sites (Table 6).

**Surface air pressure:** The highest correlations in 3-hr and daily timeseries among all variables are achieved for the surface air pressure (r > 0.9 at all sites; Table 6), while MBE reflects the elevation difference between the AWS grid cell and the actual AWS. The difference can, for example, be corrected with a use of hydrostatic balance equation. Since the variability in surface air pressure ($SP$) is associated with the atmospheric circulation at synoptic scales, it is not surprising that ERA-Interim and downscaled timeseries are highly correlated with the observed ones.

**Precipitation:** The timing and magnitude of precipitation events are generally well captured by both WRF and ERA-Interim (not shown). ERA-Interim and mod$_{7.5}$ generally overestimate the frequency of small precipitation events, while





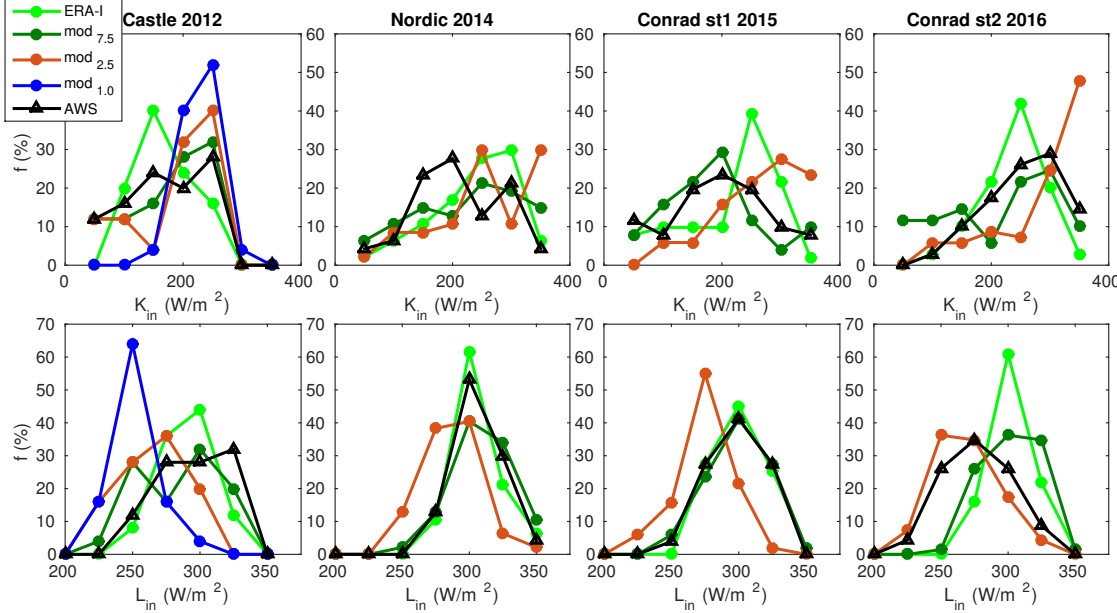

**Figure 7.** Distribution of daily $K_{in}$ ($L_{in}$) in the observation period for each site, where $f$ is number of days with given daily mean $K_{in}$ ($L_{in}$) divided by the total number of days in the observation period.

mod$_{2.5}$ underestimates (not shown). Observed rainfall rates are to be taken with caution because: (1) the rain gauge at the sites might suffer from under-sampling (as discussed in Fitzpatrick et al., 2017), and (2) the rain gauge does not adequately capture the snowfall rate that contributes to the total precipitation rate in the model - instead a combination of SR50 and rain gauge measurements should be used to assess the 'observed' total precipitation.

**Wind speed and direction:** The prevailing wind direction for the AWSs is down glacier as expected for mountain glaciers that experience katabatic flow (e.g., Oerlemans, 2001). Wind directions from WRF do not resemble the observed directions (Fig. 6) leading us to conclude that the downscaled wind fields can not correctly simulate this downslope wind direction. The WRF model, regardless of the spatial resolution, underestimates the wind speed (MBE in the range from -1.1 to -2.3 m s$^{-1}$ for mod$_{2.5}$ daily wind speed; Table 6), with the largest underestimation during clear sky days with prevailing katabatic flow (not shown). This underestimation happens despite the height differences (10 m height in WRF versus 2 m at AWS) which theoretically should lead to an overestimation of wind speed rather than the underestimation. The correlations between modeled and observed daily wind speed timeseries are relatively low (r < 0.6) and for most sites not statistically significant at 5 % confidence level (Table 6). The sites with statistically significant correlation are at Castle in 2012 and Conrad in 2016 in the accumulation area (Table 6). For Castle 2012, mod$_{1.0}$ simulates higher wind speeds than other model runs and, in part, overestimates the observed wind speed. These higher wind speeds, however, are not associated with the downslope wind direction (Fig. 6). A closer resemblance (< 1 m s$^{-1}$ difference)



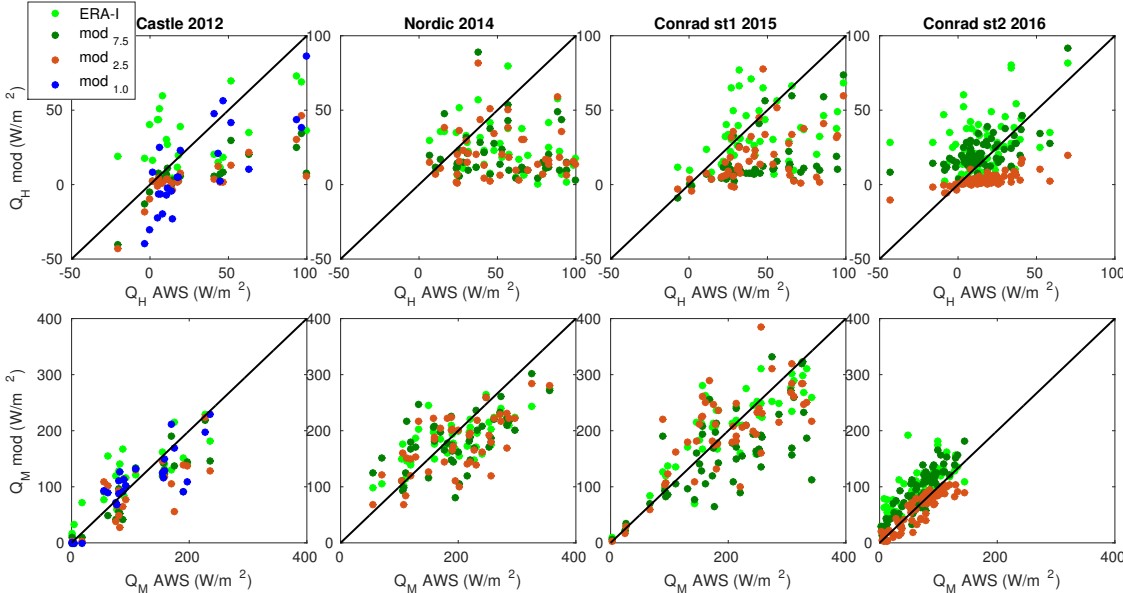

**Figure 8.** Modeled versus AWS-derived daily mean sensible heat flux ($Q_H$; upper panel) and energy available for melt ($Q_M$; lower panel) at each site during the observation period. Modeled values are given for: ERA-Interim (ERA-I), WRF at 7.5 km ($mod_{7.5}$), WRF at 2.5 km ($mod_{2.5}$), and WRF at 1 km ($mod_{1.0}$). Observed albedo and roughness lengths are used in the SEB model.

between modeled and observed daily wind speed occurs during storm events associated with surface air pressure drop and precipitation (not shown).

**Incoming radiation fluxes:** $mod_{2.5}$ consistently overestimates the occurrence of clear sky days in the observation period (Fig. 7). For the observed clear sky days, 3-hr timeseries of $K_{in}$ and $L_{in}$ closely resemble the observed fluxes (not
shown). High correlations for 3-hr timeseries in $K_{in}$ (r > 0.75) over the observation periods are mainly due to the well captured daily cycle. For the daily timeseries, however, the correlation drops (0.3 < r < 0.7 for $mod_{2.5}$; Table 6) as the overestimation of occurrences of clear sky days becomes apparent. During synoptic storm events, associated with high precipitation totals and a drop in $SP$ at AWS, modeled 3-hr $K_{in}$ and $L_{in}$ closely resemble the observed fluxes (not shown). Successfully capturing the generation of local convective clouds, on the other hand, seems to represent the
main challenge for WRF. This overall pattern of $mod_{2.5}$ performance yields an overestimated seasonal $K_{in}$ (MBE in the range from 14 to 67 W m$^{-2}$ across the sites; Table 6) and an underestimated seasonal $L_{in}$ (MBE in the range from -29 to -10 W m$^{-2}$ across the sites; Table 6). Similar biases are also evident in $mod_{1.0}$ for Castle Creek sites. To test whether the overestimation of seasonal $K_{in}$ and underestimation of seasonal $L_{in}$ in $mod_{2.5}$ is caused by the selection of grid cell, we evaluate the performance of $mod_{2.5}$ output for the surrounding eight grid cells to the AWS grid cell.
We find that, while the correlation varies depending on the choice of the grid cell, the overall pattern of positive MBE for $K_{in}$ and negative MBE for $L_{in}$, at each site, remains unchanged. Furthermore, MBE between daily $K_{in}$ timeseries





**Table 7.** Evaluation results (MBE and r) for daily timeseries of $K_{in}$ and $L_{in}$ between AWS and each of the following: ERA-Interim, $\text{mod}_{7.5}$, $\text{mod}_{2.5}$, and $\text{mod}_{2.5mean}$, which is $\text{mod}_{2.5}$ output averaged over the nine grid cells centered at the AWS grid cell. Bold r values are correlations significantly different from zero at 0.05 significance level.

| Site | ERA-I | $\text{mod}_{7.5}$ | $\text{mod}_{2.5}$ | $\text{mod}_{2.5mean}$ | ERA-I | $\text{mod}_{7.5}$ | $\text{mod}_{2.5}$ | $\text{mod}_{2.5mean}$ |
|---|---|---|---|---|---|---|---|---|
| | $K_{in}$ **MBE** (W m$^{-2}$) | | | | $L_{in}$ **MBE** (W m$^{-2}$) | | | |
| Castle 2010 | 13.9 | -45.1 | 13.7 | 20.8 | -0.3 | 11.0 | -24.5 | -21.5 |
| Castle 2012 | -1.5 | 11.1 | 20.4 | 22.8 | -1.9 | -8.5 | -28.8 | -22.5 |
| Nordic 2014 | 26.3 | 10.5 | 37.9 | 47.9 | 1.4 | 4.3 | -18.9 | -15.8 |
| Conrad 2015 st1 | 22.9 | -9.1 | 65.8 | 67.2 | -1.6 | 0.1 | -26.2 | -23.5 |
| Conrad 2016 st1 | -3.2 | -35.1 | 53.8 | 57.8 | 3.2 | 1.9 | -24.1 | -23.1 |
| Conrad 2016 st2 | -16.6 | -41.8 | 38.8 | 34.2 | 24.4 | 24.3 | -10.0 | -6.0 |
| | $K_{in}$ **r** | | | | $L_{in}$ **r** | | | |
| Castle 2010 | 0.53 | **0.63** | **0.60** | **0.65** | **0.73** | 0.55 | 0.49 | 0.47 |
| Castle 2012 | **0.80** | **0.49** | 0.38 | 0.40 | **0.73** | **0.56** | 0.37 | **0.44** |
| Nordic 2014 | **0.92** | **0.71** | **0.68** | **0.71** | **0.78** | **0.65** | **0.50** | **0.51** |
| Conrad 2015 st1 | **0.85** | **0.67** | **0.63** | **0.75** | **0.85** | **0.67** | **0.57** | **0.69** |
| Conrad 2016 st1 | **0.77** | **0.58** | **0.46** | **0.50** | **0.62** | **0.58** | **0.27** | **0.37** |
| Conrad 2016 st2 | **0.74** | **0.46** | **0.32** | **0.42** | **0.49** | **0.57** | **0.36** | **0.32** |

from AWS and $\text{mod}_{2.5mean}$, i.e. $\text{mod}_{2.5}$ output averaged over the nine grid cells centered at the AWS grid cell, is close to the MBE between AWS and the original $\text{mod}_{2.5}$ output (Table 7). Effectively, the fluxes from $\text{mod}_{2.5mean}$ represent the average fluxes over the same gridded area as in $\text{mod}_{7.5}$ (Figures 2 - 3). The fact that the seasonal $K_{in}$ in $\text{mod}_{2.5mean}$ is consistently larger than the one in $\text{mod}_{7.5}$ reveals that the explicitly resolved cloud convection in $\text{mod}_{2.5}$ yields different cloudiness from the parametrized convection in $\text{mod}_{7.5}$.

**Turbulent fluxes:** The simulation of turbulent fluxes, calculated by the bulk method, depends on how the model simulates wind speed, 2-m air temperature (for $Q_H$) and 2-m specific humidity (for $Q_E$). Since the wind speed is mainly underestimated and poorly correlated with observed values, the turbulent fluxes are also underestimated and poorly correlated with AWS-derived fluxes (Fig. 8). For $\text{mod}_{2.5}$ run, there is a consistent mean negative bias in $Q_H$ for all the AWS sites (MBE from -12 to 42 W m$^{-2}$; Table 6), while the correlation for daily timeseries spans from statistically insignificant negative correlations to statistically significant correlations of r = 0.77 and r = 0.76 (Table 6) for Castle 2012 and Conrad 2016 stations in the accumulation area, respectively. The underestimation of wind speed is likely the key reason for the underestimation of $Q_H$ since the mean bias in 2-m air temperatures is close to zero or slightly positive (within 2°C) across the study sites.

**Melt rates:** All sites show statistically significant correlations, at 5 % confidence level, between AWS-derived and WRF-derived daily timeseries of energy available for melt ($Q_M$; 0.60 < r < 0.89 for $\text{mod}_{2.5}$; Table 6 and Fig. 8). The results



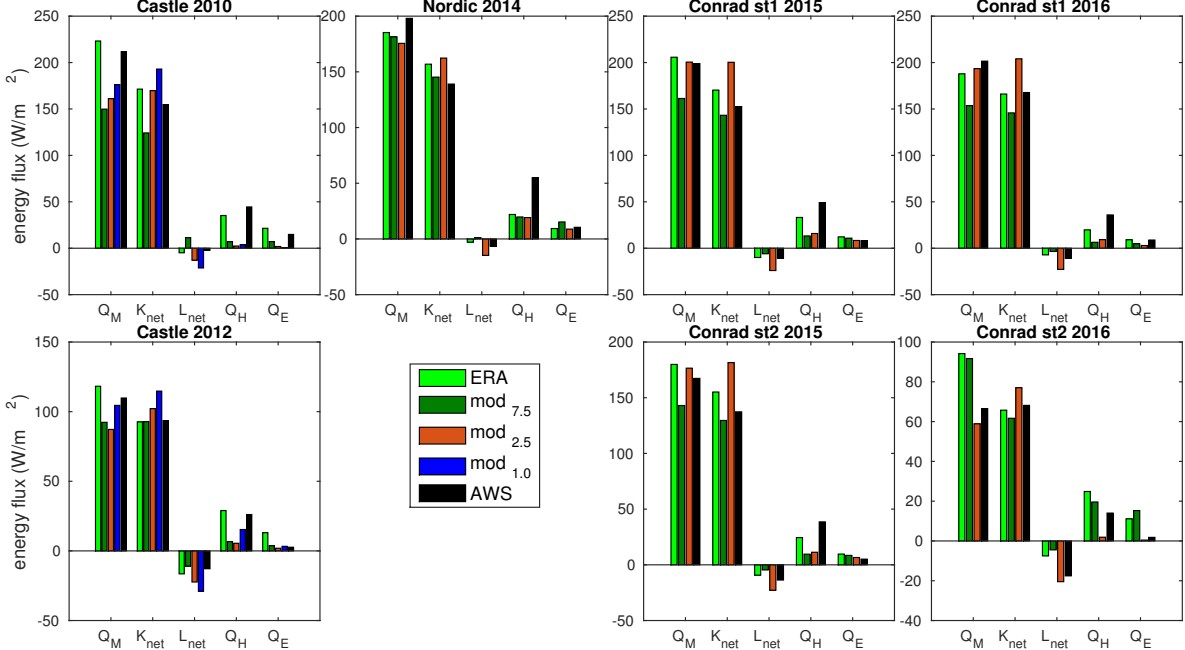

**Figure 9.** Comparison of daily mean melt energy flux ($Q_M$) and its partitioning into net shortwave radiation ($K_{net}$), net longwave radiation ($L_{net}$), sensible heat flux ($Q_H$) and latent heat flux ($Q_E$) when observed albedo and roughness lengths are used in the SEB model for each site.

here are for the observed albedo and observed roughness lengths scheme in the SEB model, but the findings are the same for the other three schemes. There is no consistent pattern in the MBE for mod$_{2.5}$ across the sites, i.e. for some sites mod$_{2.5}$ slightly overestimates $Q_M$, while for other sites there is an underestimation (Table 6). To further analyze the model performance for each site we plot the partitioning of $Q_M$, averaged over the observational period, into the net

shortwave ($K_{net}$) and net longwave ($L_{net}$) radiative fluxes, $Q_H$ and $Q_E$ (Fig. 9). At all sites, observed $K_{net}$ is the dominant contributor to the observed melt energy, followed by $Q_H$. Thus the model skill in representing the net shortwave fluxes dominates the model performance in simulating the melt rates. mod$_{2.5}$, as already mentioned, overestimates the occurrence of clear sky days, and therefore yields overestimated mean daily $K_{net}$ (up to 30 % of observed $K_{net}$) and underestimated mean daily $L_{net}$ (down to -60 % of observed $L_{net}$). Together with the underestimation of mean daily

$Q_H$ (down to -80 % of AWS-derived $Q_H$), these biases in the largest contributors to SEB act to compensate each other, resulting in a relatively successful simulation of the mean daily melt rates (difference within 10 % of AWS-derived $Q_M$) over the observation periods. Thus, despite the poorest performance of mod$_{2.5}$ run in simulating $K_{in}$ and $L_{in}$ relative to mod$_{7.5}$ and ERA-Interim, mod$_{2.5}$ gives the closest $Q_M$ to the AWS-derived values due to the compensation of biases in the SEB model. Overall, ERA-Interim displays the best skill in simulating $K_{in}$ and $Q_H$, and therefore successfully

simulates $Q_M$. Only for the AWS at the accumulation site of Conrad glacier in 2016, ERA-Interim substantially overes-



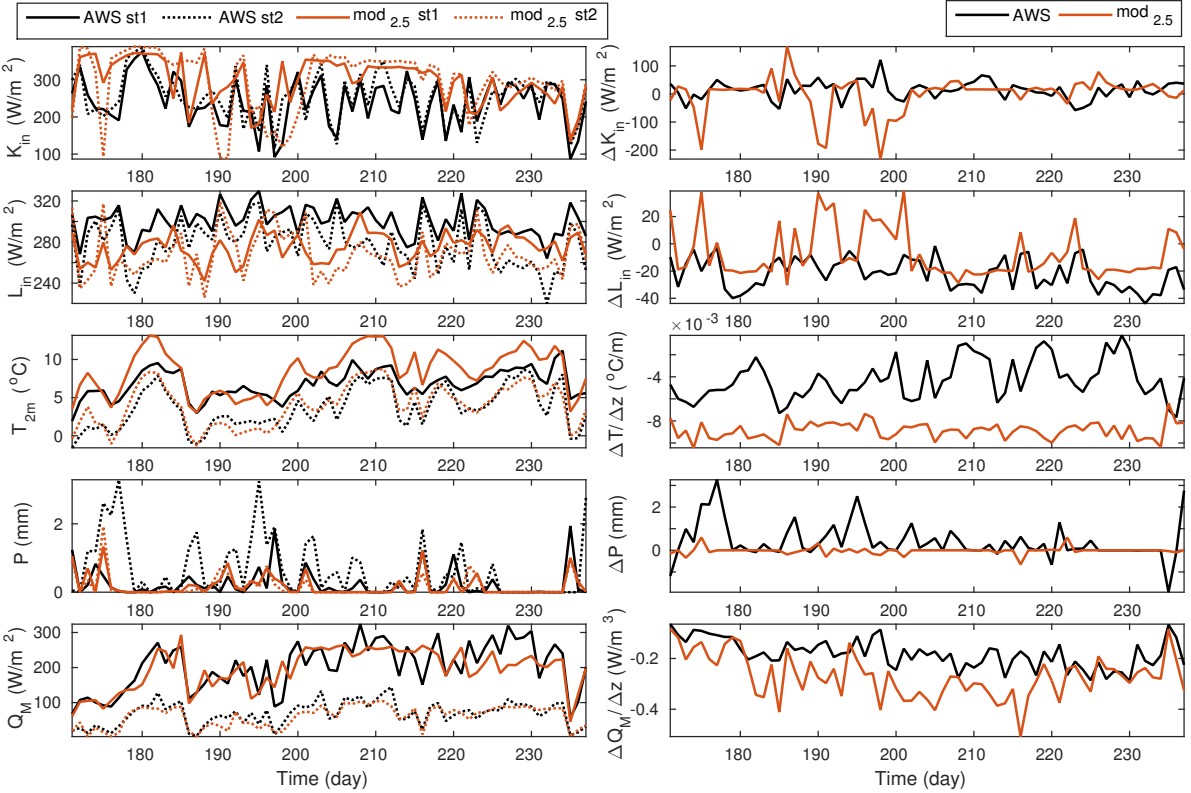

**Figure 10.** Left panel: comparison of AWS-derived versus WRF-derived ($mod_{2.5}$) daily timeseries of the selected variables from the two stations at Conrad glacier in 2016: station 1 in the ablation area, and station 2 in the accumulation area. Right panel: the difference in each variable between the two stations (daily timeseries of station 2 minus daily timeseries of station 1), as derived separately from AWS and $mod_{2.5}$.

timates $Q_M$ (up to 40 % of the AWS-derived $Q_M$) due to the overestimation of $T_{2m}$ which leads to an overestimation of $Q_H$.

## 3.2 Towards distributed melt modeling

Considering that surface melting is spatially variable, a natural extension of our analysis is to investigate how well the SEB
5  model, driven by the WRF output, can resolve this spatial variability. At Conrad glacier, two stations operated simultaneously during 2016, one station in the ablation (station 1) and other in the accumulation area (station 2), each one represented by a different grid cell in $mod_{2.5}$ (Fig. 3). For each site we plot the daily timeseries of the selected variables, observed versus $mod_{2.5}$ (Fig. 10). The SEB model here is run with the observed albedo and observed roughness lengths scheme. For each AWS-derived and WRF-derived variable we also plot the difference between the upper and lower station's timeseries (Fig. 10). Note that for
10  both $T_{2m}$ and $Q_M$, the difference is converted to a gradient, i.e. difference between the stations is divided by their elevation



difference. The results reveal smaller difference between the stations in the AWS-derived $K_{in}$ than in the WRF-derived $K_{in}$. Observed daily timeseries of $L_{in}$ show a stronger correlation between the two stations (r = 0.93), than is the case in $mod_{2.5}$ (r = 0.60). As measured at the AWSs throughout the observation period, the upper station consistently receives less longwave radiation than the lower station, while this difference is only intermittently captured in $mod_{2.5}$. The temperature gradient in

observations is consistently smaller and more variable (mean ± standard deviation: -0.43 ± 0.18 (100 m)$^{-1}$) than in $mod_{2.5}$ (-0.89 ± 0.08 (100 m)$^{-1}$). The observed temperature gradient co-varies with the wind speed measured at the lower station (r = 0.79; not shown), a pattern indicative of advective cooling during a katabatic flow. This pattern is not captured in $mod_{2.5}$. About 90 % of the time, the upper station receives higher precipitation rates than the lower station. However, the precipitation increase with elevation is replicated for less than 30 % of the time in $mod_{2.5}$. In addition, modeled and observed precipitation

display stronger correlation at the lower station (r = 0.64) than at the upper station (r = 0.24). At the lower (upper) station, the correlation between the AWS-derived $Q_M$ and $Q_H$ is 0.68 (0.77), while the correlation between the AWS-derived $Q_M$ and $T_{2m}$ is 0.74 (0.80). The equivalent correlations in $mod_{2.5}$ at the lower (upper) station are 0.22 (0.52), and 0.63 (0.85). The better model skill in downscaling wind speeds and temperature at the upper station yields a higher correlation between WRF-derived and AWS-derived $Q_M$ (r = 0.88) relative to the lower station (r = 0.73). The variability in the AWS-derived melt

gradient ($\Delta Q_M / \Delta z$) is mainly driven by temperature variability observed at the upper station (r = 0.64). Thus there is a pattern here that reveals a higher success in $mod_{2.5}$ simulations of temperature, wind speed, and turbulent fluxes, and subsequently melt energy at the accumulation site than at the ablation site.

### 3.3 Cumulative ablation and surface height changes

Figure 11 shows the modeled cumulative melt, converted to surface lowering ($z_M$), over the observation period at each site. For

all the sites except Castle Creek, $mod_{2.5}$-derived net $z_M$ is within 10 % of the AWS-derived net $z_M$. We place less confidence in the evaluation results at the Castle Creek site because the sampling period is shorter than at other sites, and the sensors measuring radiative fluxes are older and less precise than at other sites. ERA-Interim yields the closest simulation of the cumulative melt (within 10 % difference with observed melt) at each site except the accumulation site at Conrad 2016 (up to 40 % difference). Similar results are found when prescribed albedo and roughness lengths are used in the SEB model (not shown).

Next we look at the net surface height changes ($z_{net}$) that, in addition to melting, take into account the sublimation and fresh-snow accumulation. We compare the modeled $z_{net}$ to those measured by sonic rangers (SR50) at each site (Fig. 11). Since the sublimation contributes less than 1 % to the net surface changes, the largest difference in the modeled versus measured $z_{net}$ comes from the fresh-snow accumulation, as is displayed at Castle 2012 and Conrad 2016 in the accumulation area. $mod_{2.5}$-derived $z_{net}$ yields a close resemblance (difference within 10 %) to the AWS-derived and measured (SR50) $z_{net}$. In fact, at the

accumulation site of Conrad 2016, $mod_{2.5}$-derived $z_{net}$ gives closer resemblance to measured $z_{net}$ than the AWS-derived $z_{net}$ does. Using the SEB model settings with prescribed albedo, the albedo is altered from its initial value to the value for fresh snow (0.8) only during the 3-hr periods with snowfall detected at AWS or modeled in WRF. For Castle 2012, using the prescribed albedo in $mod_{2.5}$ improves the skill in simulating $z_{net}$, while at Conrad 2016 it degrades the skill. At both sites, however, $mod_{2.5}$ performs better than ERA-Interim and $mod_{7.5}$ which both fail to correctly capture the snowfall events throughout the



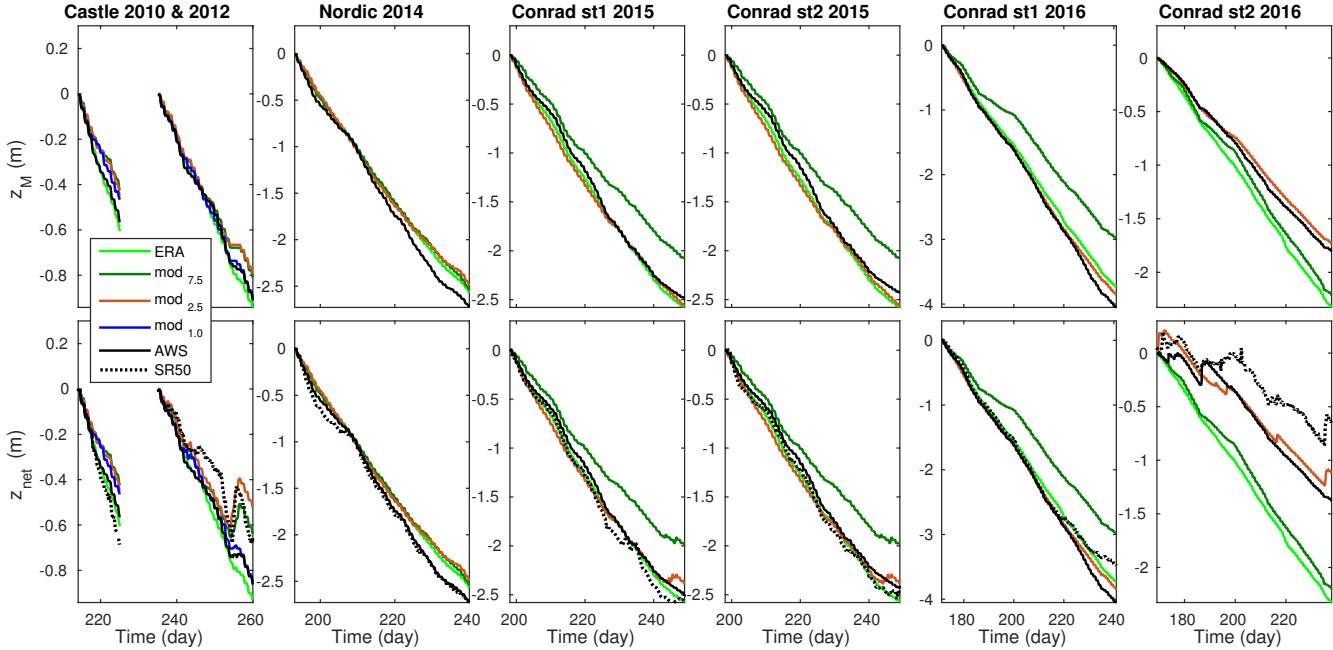

**Figure 11.** Comparison of WRF-derived versus AWS-derived cumulative melt, converted to surface lowering ($z_M$; set to negative values here), and net surface height changes ($z_{net}$) at each site over the observation period. Observed albedo and roughness lengths are used in the SEB model. Dotted black line represents measured surface height changes from the processed SR50 data for each site.

observation periods. Since the variance in $z_M$ explains the bulk of the variance in $z_{net}$, there is a close resemblance between WRF-derived and measured (SR50) $z_{net}$ for mod$_{2.5}$ regardless of how well WRF simulates the snowfall events. Nevertheless, we show here that correctly downscaling temperature and precipitation plays an important role in $z_M$ and $z_{net}$ modeling through the albedo feedback.

## 3.4 Sensitivity to spatial resolution and landcover in WRF

With the $SSC$ metric we analyze (1) the downscaling performance relative to the reference ERA-Interim simulations, and (2) whether the increase of spatial resolution (from 7.5 to 2.5 and 1 km) adds skill to the WRF model simulation. $SSC$, calculated for the variables used in the SEB model, show that the downscaling only improves ($SSC > 0$) the simulation of 2-m air temperature (Fig. 12). Inter-comparison of $SSC$ for model runs of different spatial resolution reveals the following pattern:

for the radiative fluxes ($K_{in}$ and $L_{in}$), the model run with a finer resolution consistently yields poorer skill across all the sites. Precipitation is better simulated in mod$_{2.5}$ than in mod$_{7.5}$ for all the sites except Castle 2012, while for wind speed only half of the sites are better simulated in mod$_{2.5}$. The effect of refining the model resolution from 2.5 to 1 km, as tested on the Castle site in 2010 and 2012, does not improve the skill for any variable. While one might expect a katabatic flow to be better resolved at 1 km resolution, this is certainly not the case for our sites where the $SSC$ shows more negative values for mod$_{1.0}$ than for





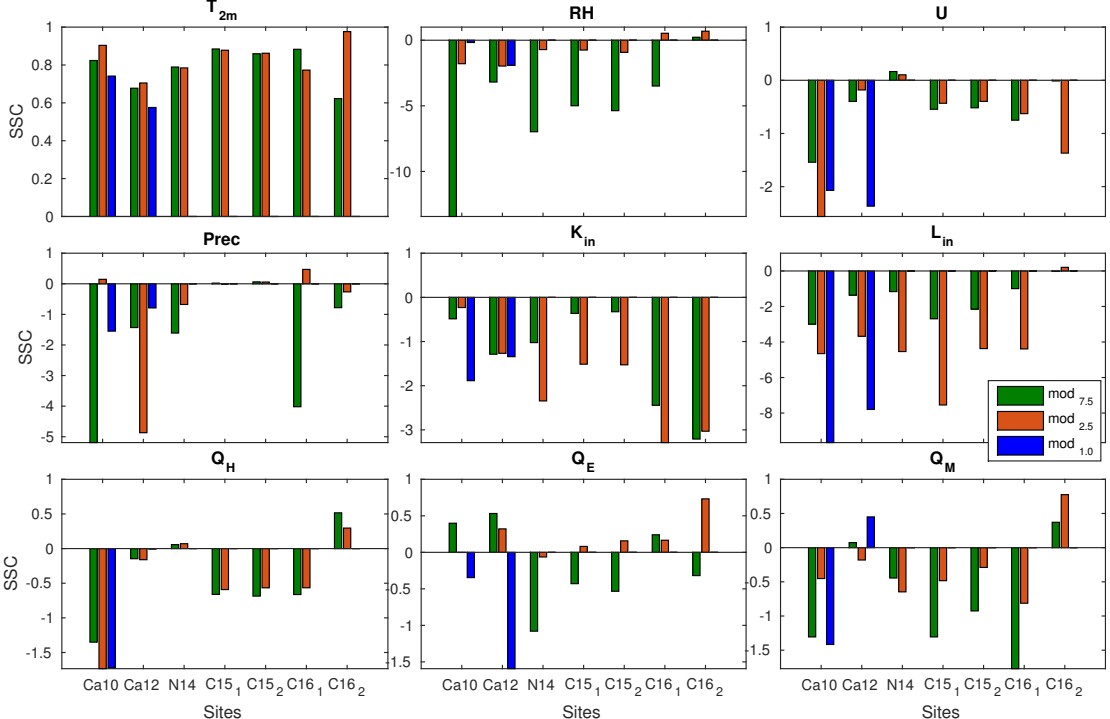

**Figure 12.** Skill Scores ($SSC$) calculated for daily timeseries of the selected variables at each study site. The sites are abbreviated: Castle Creek in 2010 (Ca10), Castle Creek in 2012 (Ca12), Nordic in 2014 (N14), Conrad in 2015 st1 (C15$_1$), Conrad in 2015 st2 (C15$_2$), Conrad in 2016 st1 (C16$_1$), and Conrad in 2016 st2 (C16$_2$).

mod$_{7.5}$. Finally, at most sites, mod$_{2.5}$ has a better skill than mod$_{7.5}$ in simulating $Q_M$. As previously shown, the success in simulating $Q_M$ depends on the bias compensations in the components of the SEB. The effect of bias cancellation works the best in the mod$_{2.5}$.

To analyze the effect of landcover on the WRF performance, we look into the WRF output of the grid cell with snow/ice 5  landcover, labeled as mod*$_{2.5}$ and mod*$_{1.0}$, that is spatially the closest to the grid cell representing the AWS. In addition, for Nordic site in 2014, we extract the model output at 2.5 km resolution where the landcover category of the AWS grid cell was manually altered from the bare tundra to snow/ice (mod$_{2.5-modified}$). By doing so we altered the albedo in WRF for that grid cell from 0.18 (bare tundra) to 0.67 (snow/ice). The albedo in the SEB model, however, remains unaltered, i.e. set to the observed daily albedo at the AWS. The results of these runs reveal that the sensitivity to the landcover alteration, as tested for 10  the Nordic site, is negligible, i.e. mod$_{2.5-modified}$ produces a small increase (up to 1 %) in each energy flux of the SEB relative to mod$_{2.5}$. On the other hand, mod*$_{2.5}$ and mod*$_{1.0}$ yield larger SEB differences from their original runs at each site (up to 10 % difference). This finding is not surprising considering that mod*$_{2.5}$ and mod*$_{1.0}$ give output for a grid cell of different elevation, location and landcover than the grid cell representing the AWS site.





### 3.5 Sensitivity to parameters in the SEB model

Here we analyze the sensitivity of modeled seasonal $Q_M$ to the four schemes for albedo and roughness lengths in the SEB model. Results reveal that switching between the two albedo schemes has more effect on altering $Q_M$ than switching between the roughness length schemes, as expected due to the dominant effect of $K_{net}$ on available energy for melting. As shown earlier, the largest sensitivity of daily mean $Q_M$ to the albedo setting is at the sites with observed snowfall during the observation period. At Castle 2012, the difference between prescribed vs observed albedo yields a difference of 20 % in the seasonal $Q_M$, while for Conrad 2016 (in the accumulation area) the difference is 30 %. The difference at all other sites is within 10 %. The results confirm the pattern where the larger the difference between the prescribed and observed albedo (Fig. 4), the larger the difference in seasonal $Q_M$. The effect of different roughness length schemes on altering $Q_H$, and consequently on altering $Q_M$, is small because (1) $z_{0v}$ is of the same order of magnitude ($10^{-3}$ m) in both schemes (Table 5), and (2) $z_{0T}$ and $z_{0q}$, derived through the surface renewal model (Andreas, 1987), agree within the order of magnitude with the observed values, i.e. values derived from the eddy-covariance measurements (Fitzpatrick et al., 2017; Radić et al., 2017). We conclude that the choice of roughness lengths, as long as their order of magnitude agrees with the observed one at these sites ($10^{-3}$ m for $z_{0v}$, and $10^{-5}$ m for $z_{0T}$ and $z_{0q}$), has a small effect on the seasonal $Q_H$ ($< 5$ % difference) and consequently negligible effect on the seasonal $Q_M$ ($< 1$ % difference).

### 3.6 SEB model versus PDD model performance

The simple PDD model forced by $mod_{2.5}$ 2-m air temperature, using the calibrated melt factors ($f_m$, Table 5), simulates the net melt ($z_M$) over the whole observation period within 10 % difference from the AWS-derived $z_M$ using the SEB model (Fig. 13). Despite the good performance of the PDD model, its major limitation is the required calibration of the melt factors. When altering the calibrated value of $f_m$ by $\pm$ 1 mm w.e. day$^{-1}$ $^\circ$C$^{-1}$ at each site, net $z_M$ is changed by up to 40 % from its original value (Fig. 13). Note that the perturbation in the melt factor of $\pm$ 1 mm w.e. day$^{-1}$ $^\circ$C$^{-1}$ is relatively small considering that, for a set of neighboring glaciers in the region, reported glacier-specific $f_m$ for ice and for snow are in the range 4.0 - 9.7 mm w.e. day$^{-1}$ $^\circ$C$^{-1}$ and 2.4 - 6.6 mm w.e. day$^{-1}$ $^\circ$C$^{-1}$, respectively (Radić and Hock, 2011). In addition, as has been previously noted (see Hock, 2005), it is questionable whether the melt factor for the same location can be treated as a constant in time. In our study, the melt factor for ice differs by more than 1 mm w.e. day$^{-1}$ $^\circ$C$^{-1}$ when calibrated for, approximately, the same site in two different seasons (Castle site for 2010 and 2012, and Conrad station 1 in 2015 and station 2 in 2016; Table 5).

## 4 Discussion

Several recent studies showed a successful application of WRF in dynamically downscaling meteorological fields needed to force a SEB model for glacier surfaces. Here we focused on evaluating WRF in simulating meteorological variables and fluxes needed to force a single-point SEB model at three mountain glaciers in the interior mountains of British Columbia. As reference data for model evaluation, we used observations from the AWSs operating intermittently at these glaciers in summers of 2010-





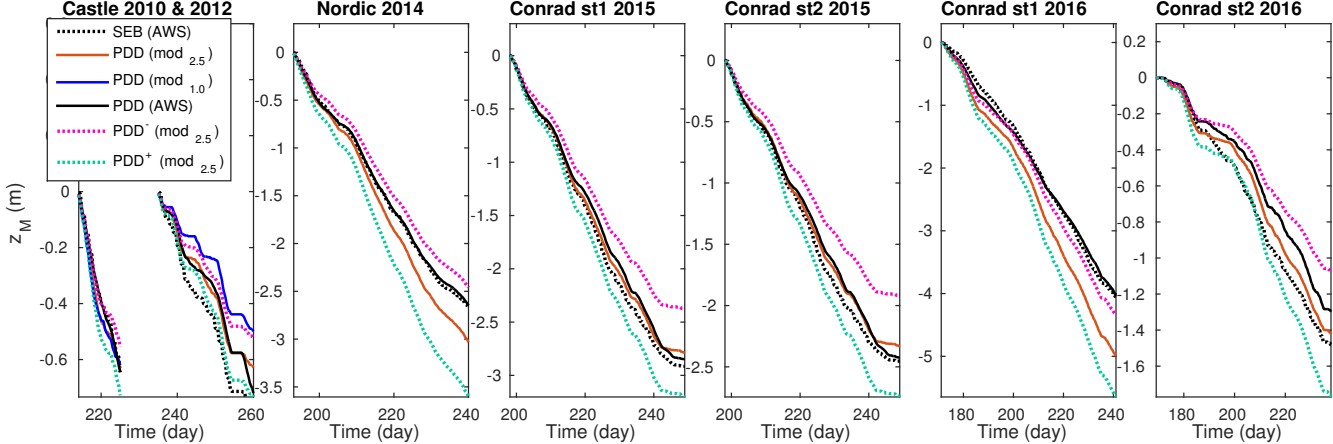

**Figure 13.** Cumulative melt converted to surface lowering ($z_M$) as simulated by the simple PDD model and by the SEB model (with observed albedo and roughness lengths) for each site. The PDD model is forced by 2-m air temperature from AWS and from WRF at 2.5 km and 1 km spatial resolution. Results of the sensitivity test where the calibrated site-specific melt factor in the PDD model is perturbed by $\pm$ 1 mm w.e. day$^{-1}$ °C$^{-1}$ are shown for mod$_{2.5}$ (PDD$^-$ and PDD$^+$).

2016. The ERA-Interim reanalysis was dynamically downscaled with WRF in three domains with resolutions of 7.5 km, 2.5 km, and 1 km, where the inner-most domain of 1 km resolution was only used for one glacier. The range of spatial resolutions allowed us to analyze the sensitivity of the downscaled variables to the choice of spatial resolution. We also analyzed the sensitivity of modeled melt rates to alternating the landcover of the grid cell representative of AWS in WRF, and to the choice

of parameterization schemes for albedo and surface roughness in the SEB model.

With the exception of near-surface air temperature, all meteorological variables measured by AWS in the ablation areas of the glaciers are better or equally well simulated by ERA-Interim relative to WRF. In contrast, WRF at 2.5 km resolution performs best for simulating daily melt rates for the site in the accumulation area. The better performance of ERA-Interim over the WRF model is, in part, expected since data assimilation is incorporated in the reanalysis model, while the WRF model does

not include any nudging to available in-situ and/or remote sensing observations for the domain.

WRF at 2.5 km resolution overestimates the frequency of clear sky days over the observation periods, which leads to an overestimation of the seasonal incoming shortwave radiation ($K_{in}$), and an underestimation of the seasonal incoming longwave radiation ($L_{in}$). Both ERA-Interim and WRF at 7.5 km resolution show better simulation (e.g. smaller MBE and/or higher correlation) of radiative fluxes than WRF at 2.5 km resolution. These results corroborate previous studies in demonstrating that

the overestimation of $K_{in}$ in the finest resolution WRF run indicates a problem with convective cloud simulation over complex terrain (e.g., Claremar et al., 2012; Collier et al., 2013; Schanke et al., 2015). An increase in the complexity of the topography affects the cloudiness. If this effect is not sufficiently well described by the model resolution, it may induce more errors in the cloud cover relative to a coarser resolution. Furthermore, we do not use a cumulus parameterization in the innermost domain (2.5 and 1 km) assuming that cumulus convection is explicitly resolved. However, our results show that, over the same-sized





gridded domain, the cumulus convection in $\text{mod}_{2.5}$ gives different results for $K_{in}$ and $L_{in}$ than the cumulus parameterization in $\text{mod}_{7.5}$, the latter giving better resemblance with AWS-derived radiative fluxes. We conclude that the WRF resolution of 2.5 and even 1 km might not be fine enough to correctly resolve the cumulus convection in this complex terrain. Some previous studies indicated that a grid spacing on the order of 100 m (Bryan et al., 2003; Petch, 2006) or even 10 m (Craig and Doernbrack,

2008) is needed to capture the dominant length scales of moist cumulus convection.

The finer WRF resolution (2.5 km versus 7.5 km) has better skill in simulating melt rates, but this is mainly due to the cancellation of biases in the SEB. The positive bias in seasonal $K_{in}$ is compensated by the negative biases in seasonal $L_{in}$ and sensible heat fluxes ($Q_H$). The underestimation of $Q_H$ down to -80 % is due to underestimated near-surface wind speeds used in the bulk aerodynamic method. As shown for our sites in the ablation areas, WRF fails to simulate the prevailing

downslope katabatic flow and, therefore, fails to capture the adiabatic cooling in the ablation area. Because wind speed is equally important as temperature in calculating $Q_H$ via the bulk method, the poor simulation of wind speeds explains the poor performance in estimating $Q_H$. On the other hand, better WRF performance in both temperature and wind speed is found for the site in the accumulation area. While WRF at few-kilometer resolution is able to simulate katabatic or other slope winds at large outlet glaciers (> 100 km$^2$) and ice caps (Claremar et al., 2012), it fails to do so at our sites even at 1 km spatial

resolution. Considering that the smallest weather features that can be modeled by WRF are about seven times the grid spacing (Warner, 2011), the largest horizontal spacing needed to resolve katabatic wind is approximately: 0.8 km for Castle Creek, 0.3 km for Nordic, and 1 km for Conrad glacier. We therefore conclude that, for relatively small glaciers (< 7 km length along the flowline) in complex terrain, the 1 km grid spacing is not fine enough to simulate the local topographic wind effects which highly depend on local gradients in the topography that correctly captures ice/snow cover. As indicated in Claremar

et al. (2012), increasing the vertical resolution in the atmospheric boundary layer (e.g., 2 m grid in the lowest 10s of meters; Soderberg and Parmhed, 2006) might improve the simulation of near-surface wind speed and direction. It is likely that, for our sites, an increase in both horizontal and vertical resolution would be needed for better simulation of katabatics, but this would also increase computational time of the simulations. Another problem in simulating $Q_H$ is the use of bulk methods rooted in the Monin-Obukhov theory, which can be inadequate in its application to sloped glacier surfaces (e.g., Radić et al., 2017).

While we used the downscaled temperature and wind speed in the bulk method for $Q_H$, the method is based on the same theory used for parameterizing the boundary layer turbulence in the WRF model (e.g., Janjic, 1996; Nakanishi and Niino, 2009). To potentially improve the simulation of katabatics and $Q_H$ at our sites in the ablation area, without increasing the resolution in WRF or running an eddy-simulator, one could make use of the successfully downscaled variables at the accumulation area. Downscaled temperature and wind speed in the accumulation area could serve, for example, as boundary conditions

to a katabatic flow model (Prandtl, 1942) solved with an elevation-varying eddy viscosity in the flux-gradient method (e.g., Grisogono and Oerlemans, 2002).

A set of sensitivity tests showed that modifying the landcover category in WRF at 2.5 km for the grid cell representative of the AWS site has negligible effect on the downscaled variables. This sensitivity analysis, however, consisted of altering the landcover of only one grid cell in the domain, which is likely too small a perturbation in the input conditions to substantially

impact the model output. Furthermore, since the albedo in the SEB model is treated independently of WRF, the modeled



melt rates are also not affected by this alteration in the landcover. It remains to be seen, however, how important the landcover alteration would be in a coupled WRF-SEB modeling approach, as previously addressed in Mölg et al. (2012a) for Kilimanjaro. The modeled melt rates are shown to be sensitive to the albedo scheme in the SEB model, indicating the importance of (i) accurate representation of albedo, and (ii) a correct simulation of snowfall events throughout the ablation season. While the

first point can be addressed by incorporating a better albedo model (e.g., Schmidt et al., 2017), simulating snowfall, and precipitation in general, remains a major challenge for WRF. Underestimation of frequency and intensity of precipitation is consistent with an overestimation of $K_{in}$. As pointed out by Collier et al. (2013), this performance may reflect errors in the forcing data at the lateral boundaries and/or in the WRF resolution to fully resolve orographic uplift or microscale complex flow features that affect precipitation at the sites. The modeled melt rates are shown to be insensitive to the choice of roughness

lengths as long as the roughness length for momentum ($z_{0v}$) agrees with the order of magnitude of its observed value and the scalar roughness lengths ($z_{0T}$ and $z_{0q}$) are calculated from the surface renewal model of Andreas (1987). The value of $z_{0v}$ used here ($10^{-3}$ m) is commonly assigned for ablation area on mid-latitude glaciers (e.g., Cuffey and Paterson, 2010).

Finally, we discuss the implications of our results for the potential use of distributed SEB models to obtain spatially resolved ablation rates at a glacier of interest. Our results demonstrate that using dynamically downscaled variables at few-kilometer

spatial resolution might not satisfy the needs for rigorous distributed SEB modeling. Firstly, a sub-kilometer spatial resolution (e.g., 10-100 m) would be needed to successfully resolve cumulus convection and radiative fluxes. If more than one melt season is to be simulated, however, running WRF at the sub-kilometer grid would be computationally expensive. For comparison, it took us approximately half a day to complete a one day $\text{mod}_{1.0}$ simulation on the supercomputer. Secondly, considering the success of ERA-Interim in simulating the radiative fluxes, we advocate for using the radiative fluxes directly from the reanalysis

datasets in the SEB models. Instead of downscaling $K_{in}$, the spatial variability in $K_{net}$ could be captured through a spatially varying albedo along the glacier surface. In particular, the albedo model would need to capture, in time and space, the surface transition from snow to ice, and the snowfall rates. To do so, both temperature and precipitation fields would need to be dynamically downscaled to at least 2-3 km resolution. Depending on the desired accuracy and resolution of the distributed SEB model, these fields could further be downscaled via statistical methods that use, for example, assigned or modeled lapse

rates and precipitation gradients. Alternatively, further downscaling could be achieved by running simplified physics-based models, for example, an aforementioned katabatic flow model to obtain temperature and wind profiles in the surface boundary layer, and a model for orographic precipitation (e.g., Smith and Barstad, 2004; Jarosch et al., 2010).

## 5  Conclusions

The main goal of this study was to investigate the performance of a dynamical downscaling approach to derive meteorological

fields needed to force a SEB model at three mountain glaciers in the interior of British Columbia. We showed that, with the exception of near-surface air temperature, all meteorological variables are better simulated by ERA-Interim at 80 km horizontal resolution than by WRF at 2.5 and 1 km resolution. ERA-Interim was shown to adequately simulate the glacier-scale radiative fluxes that are the dominant drivers of seasonal melt rates. WRF, on the other hand, was somewhat successful in simulating

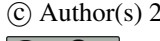



the radiative fluxes but unsuccessful in simulating the near-surface wind speed and turbulent heat fluxes. The cancellation of biases in the most relevant components of SEB was responsible for the relative success of WRF at 2.5 km in simulating the seasonal melt rates, with modeled values falling within 10 % of the AWS-derived ones. Considering the limitations with the WRF downscaling and relatively good performance of ERA-Interim, we advocate for the SEB modeling directly forced with

reanalysis datasets for the past and present climate, and from GCMs for future climate scenarios. Those GCMs, however, need to be evaluated against the reanalysis products prior to their applications in the projections. Temperature downscaling remains to be an important step for the simulation of turbulent fluxes, especially if one is interested in capturing the variability in melt rates throughout the ablation season. More importantly, however, we found that the WRF failure to resolve the katabatic flow in the ablation area of the relatively small glaciers (< 7 km length along the flowine) deteriorates the simulation of turbulent

heat fluxes even when near-surface air temperature is successfully downscaled. A way forward, as we see it, is to use the air temperature and wind fields, downscaled to the glacier accumulation area, as boundary conditions to a locally nested katabatic flow model.

*Code and data availability.* This study is based on the output from a Weather Research and Forecasting (WRF) model, version 3.8.1, available for download at http://www2.mmm.ucar.edu/wrf/users/downloads.html. The AWS data from Castle Creek and Nordic glaciers are

available upon request from the corresponding author. Data from Conrad sites will not be released prior their publication as part of a separate study. The raw AWS data are not currently in a standard format, and the metadata are not fully digitized.

*Author contributions.* M. A. Tessema preformed the WRF model setup and runs, and contributed to the analysis of WRF output and manuscript write-up. V. Radić participated in and financed the field campaigns, developed the methodology, contributed to the analysis of WRF output, and wrote the initial version of the manuscript. B. Menounos helped with financing the field campaigns, helped with set-

ting up the WRF runs and contributed to the manuscript refinement. N. Fitzpatrick led the field campaigns on Nordic and Conrad glaciers, collected and processed the AWS data, and contributed to the manuscript refinement.

*Competing interests.* The authors declare that they have no conflict of interest.

*Acknowledgements.* Funding supporting this study was provided through the Natural Sciences and Engineering Research Council (NSERC) of Canada (Discovery grants to V. Radić and B. Menounos). Our meteorological equipment were supported by NSERC Research Tools

and Instruments grant and Canada Foundation for Innovation grant (V. Radić). WRF runs are facilitated by the Compute Canada WestGrid super-computer. Special thanks to Emily Collier for the guidance with the WRF setup and to Charles Krzysik for the IT support. Gwenn Flowers, Douw Steyn and Roland Stull are thanked for discussions and constructive criticism on the initial version of the manuscript.




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
