# Peer review of "Evaluation of dynamically downscaled near-surface mass and energy fluxes for three mountain glaciers, British Columbia, Canada"

_The Cryosphere, 2018_

## Referee Comment (RC1) · Anonymous Referee #1 · 13 Oct 2018

**Review of "Evaluation of dynamically downscaled near-surface mass and energy fluxes for three mountain glaciers, British Columbia, Canada"**

**General remarks**

Tessema et al. (2018) compare meteorological fields simulated using the atmospheric model WRF with *in situ* measurements for three glaciers in British Columbia, Canada, and assess differences in point simulations of glacier surface energy and mass balance using these two datasets as forcing. The authors examine the impact of

model resolution and correctly specifying the underlying land surface type on their analysis, as well as compare with the positive degree-day method. Based on this work, the authors draw conclusions about the feasibility and success of using dynamical downscaling with WRF to produce forcing for glacier simulations.

This manuscript joins only a small number of other studies on this topic, and the analysis is strengthened by the availability of numerous observations for model evaluation. The manuscript is logically organized and well written, although at times unnecessarily convoluted, and the topic is well suited for The Cryosphere. However, I have a number of concerns about the numerical modelling, outlined below, that I think need to be addressed before publication. Given these issues, some of the conclusions presented in the paper are not supported by the presented analysis. Improving the study by providing more accurate atmospheric simulations will greatly strengthen its contribution to cryospheric community.

**Major comments**

1. There are two key issues with the atmospheric simulations:

a. The authors have not justified their choice of model physics. The configuration does not match any of the references on Page 6 (P6), Line 9 (L9), but in any case, those studies focused on different mountainous and climatic regions. As the authors mention, the choice of physics has a large impact on WRF results and should be optimized based on a subset of the observational data or justified in some way.

b. The land surface in the closest grid cell to the observations in the ablation zones is not classified as land ice. This inconsistency is easy to fix manually in the geo_em files before running the WRF pre-processing programs. Manual correction in the finest resolution domains (WRF 2.5 and 1.0) would appear to result in only one incorrect land-use categorization at observational points, for the southern off-glacier AWS at

Castle Creek (cf. Figure 2). Reliable conclusions about the model's ability to reproduce local meteorological conditions and katabatic flows cannot be drawn when the bottom boundary conditions are incorrectly specified, and as a result, there are no glacierized grid cells neighboring the one containing the station (WRF 1.0 at Castle Creek) or there is only a single glacierized grid cell (WRF 2.5 at Nordic Glacier).

The authors attempt to address the second issue by manually changing the land-surface type in the grid point containing the AWS at Nordic Glacier. However, this is the smallest glacier studied and may represent an underestimate of the impact of atmosphere-glacier feedbacks on the presented results. In addition, the horizontal resolution of the finest domain (WRF 2.5) is not well suited for this study site, as the authors acknowledge on P27, L15. For these reasons, I think the simulations should be repeated with accurate bottom boundary conditions (see minor comment 2 about SST). This change would also help to streamline and simplify the manuscript (i.e., the authors could remove P6, L29-34; P13, L17-26; P14, L24-29; Section 3.4).

Please see my minor comments for a few more questions and concerns about the WRF simulations.

2. The approach to the glacier simulations underutilizes the information provided by WRF, perhaps due to the incorrect land-surface categorization. For example, for most of the SEB simulations presented in the paper, daily mean albedo is specified from observations for both AWS- and WRF- forced runs (e.g., P20, L1; P21, L8) rather than using the simulated WRF value (which should be optimized in the source code, see minor comment 9). They use a calibrated value for fresh snow density and apply a temperature threshold for determining the frozen fraction of precipitation, however both of these fields are available from the microphysics scheme. This approach, in particular the albedo treatment, makes the comparison between the two forcing datasets less

informative than it could be.

**Minor comments**

1. P6, L7: Why did the authors create a new set of domains for the simulation down to 1-km grid spacing at Castle Creek? Other options would be to nest a fourth domain or use the WRF program ndown to force two separate D3s of 2.5-km and 1-km grid spacing. The latter would be the most consistent in terms of lateral boundary information, and both would be more numerically efficient.

2. P6, L15: Why was SST kept constant and how does this impact the simulations, some of which exceed two months in length? This time-varying field is provided by ERA-Interim and is easily incorporated into the simulations using the wrflowinp bottom boundary updates.

3. P6, L24: Please provide the exact spin-up time and a reason for this choice.

4. P7, Table 3:

a. Are the timesteps correct? If yes, is the model solution stable and physical with a timestep of 20 times the grid spacing at Castle Creek?

b. In complex terrain, diffusion should be computed in physical space (diff_opt set to 2) for more accurate results where coordinate surfaces are sloped.

5. P8, Figure 2: The authors state that they updated the land-use data using ESA CCI, however certain areas that appear to be at least 506. P9/10: The paragraph explaining the bulk aerodynamic method could be removed, since it is well established and the reader is referred to Fitzpatrick et al. (2017).

7. P10, L15-16: What ice/snow albedos are supported by the measurements?

8. P16: I suggest showing the comparison for precipitation, as it plays a role later in the modelled surface height changes.

9. P24, L8: The albedo for glacierized grid cells is a prognostic variable in WRF. The default lower bound (variable ALBICE in phys/module_sf_noahmp_glacier.F) is set unrealistically high at 0.67 and should be changed to a value consistent with bare ice for more realistic simulations and atmospheric feedbacks.

10. P26, L2: Please rephrase to match what is described in the methods.

11.  P28, L17: Further optimization may be possible.  For comparison, we run a three-domain configuration down to sub-kilometer grid spacing with dimensions exceeding 300x300x50 and are able to complete more than 20 simulation days in one day of wall-time on 500 processors.

**Technical comments**
1. I suggest changing "near-surface" to "surface" in the title, as only surface mass and energy fluxes are considered.
2. P1, L5: for clarity, I suggest changing "nested within the ERA-Interim" to "forced at its lateral boundaries by ERA-Interim."
3. P1, L6: change "spatial resolution" to "grid spacing."
4.  Section 2.1: I suggest referencing Figures 2 and 3 where applicable to make it easier for the reader to follow the station locations.
5. P6, L2: "advanced research version of the WRF model."
6. Throughout the paper, please change "(see text)" to refer to the relevant section.
7.  P14, L28 and elsewhere: I think the phrase "(not shown)" is overused in the manuscript.  I suggest removing some of the statements if they are not important or introducing supplementary material.
8. P27, L4-5: This sentence appears nearly word-for-word in Collier et al. (2013).

**Tables  Figures**
Figure 1: I suggest adding shaded model topography.
Figures 2 and 3: Please label the axes or provide a scale. For Figure 3, is there a pink triangle?
Table 3: Please provide the grid dimensions.

---

## Referee Comment (RC2) · M. Lehning (Referee) · 1 Nov 2018

The paper uses the Weather and Research Forecasting Model (WRF) to dynamically downscale weather information from ERA-Interim global reanalysis to investigate potential improvement of the surface mass- and energy balances of three mountain glaciers in BC, Canada.

The set-up of WRF for such a downscaling analysis, the collection of related field data and the analysis of model in comparison to the measurements is a lot of work and a

great effort. This is fully acknowledged but at the same time, the set-up of the study is such that the negative results (no improvement through WRF) are no surprise. The study is heavily outdated in a time when weather models run at 2 km resolution operationally (e.g. Cosmo in Switzerland) and WRF downscaling work is done at the resolution of tens of meters (Gerber et al., 2018) and not kilometers as in the submitted paper. A study that has just been published and uses a 30 m resolution to generate meteorological input for glacier energy and mass balance in the Himalayas is Stigter et al. (2018) and is showing the state of the art in the field and in fact documents that WRF is able to successfully generate useful local weather input.

From our own early work (Lehning et al. 2008; Raderschall et al., ), we know that it needs a very high resolution of below 50 m to approximate wind fields in complex terrain. And if you have correct high-resolution wind fields, you can describe boundary layer processes from snow deposition (Mott and Lehning, 2010) to snow ablation (Mott et al., 2011). I would go as far as to say that the dynamical downscaling of first flow and then full weather has been initiated with these ARPS based studies and the methodology has immediately also been applied to glaciers (Mott et al., 2008; Dadic et al., 2010) including the Mölg and Kaser study (Mölg and Kaser, 2011) mentioned in the paper. Therefore, if you do a downscaling study that will not reproduce the local wind fields correctly (in this case katabatic flows), you cannot expect to see much improvement over a re-analysis on local snow mass balance estimates. Now, if this was the first attempt and all that is currently possible, then this could still be interesting. But if much older and current studies (such as the ones mentioned above) already have pushed the limits way beyond the setup of this study and state of the art downscaling in much higher resolution shows that you can even simulate individual eddies (Gerber et al., 2017) then the results as presented in the paper do not add to our scientific body of knowledge.

Let me please emphasize that I am not trying to reject the paper because our own work and other work as discussed above did not get cited. This may be seen as an omission

but is not even necessary or could be fixed easily. The main point is really that the results do not allow to gain new insight and give a wrong impression on the usefulness of dynamical downscaling because the study had the wrong/outdated design despite all the good work that has been done in the execution. I would also fully be in favor of publishing negative results but not if the negative results are the consequence of an inadequate set-up such as in this study. This is unfortunate, as the paper is really well written and nicely illustrated.

In addition to the general comments , I have also one additional major set-up problem, which is the arbitrary switch-off of the cumulus convection scheme, while it is quite clear that convection will be insufficiently resolved at a 2.5 km grid resolution. Again, it then no surprise that precipitation simulations have a large error.

One final major point, which can either be a typo or a serious misconception is the statement on p. 27 l. 10 when the authors talk about adiabatic cooling in the katabatic wind on the glacier. Of course, descending air masses warm by an adiabatic process.

Some detailed comments:

p. 5 l. 6: Roughness lengths should be consistent with model resolution as sub-grid topography needs to be represented by the roughness

p. 5 l.8: Why did you not use a precipitation lapse rate?

p. 9 Eq. 2: I don't know why there is $p/p_0$ in this equation. This does not come from the original derivation of the bulk formula (see e.g. textbooks by Brutsaert or Stull) and the influence of air pressure is already there via the air density.

p. 19 l.11: There is some physical argument why roughness for momentum could be different from roughness of scalars (but empirical evidence is missing, see e.g. Schlögl et al., 2017). However, assuming two different roughness lengths for moisture and temperature has no theoretical justification

p. 13 l.3: You could diagnose local stability (at least over a melting surface – not

sure you had a surface temperature measurement) and then use an adequate stability correction.

References: Dadic, R., R. Mott, M. Lehning, and P. Burlando (2010), Wind influence on snow depth distribution and accumulation over glaciers, J. Geophys. Res., 115, F01012, doi:10.1029/2009JF001261.

Gerber, F., Besic, N., Sharma, V., Mott, R., Daniels, M., Gabella, M., Berne, A., Germann, U., and Lehning, M.: Spatial variability in snow precipitation and accumulation in COSMO–WRF simulations and radar estimations over complex terrain, The Cryosphere, 12, 3137-3160, https://doi.org/10.5194/tc-12-3137-2018, 2018.

F. Gerber, M. Lehning, S. W. Hoch, et R. Mott, Âń A close-ridge small-scale atmospheric flow field and its influence on snow accumulation Âż, Journal of Geophysical Research:Atmospheres, vol. 122, no 15, p. 18. 7737-7754, 2017.

Lehning, M., Löwe, H., Ryser, M., Raderschall, N., 2008. Inhomogeneous precipitation distribution and snow transport in steep terrain, Water Resour. Res., 44, W07404, doi:10.1029/2007WR006545.

Mölg, T. and Kaser, G.: A new approach to resolving climate-cryosphere relations: Downscaling slimate dynamics to glacier-scale mass and energy balance without statistical scale linking, Journal ofGeophysical research, 116, 779–795, https://doi.org/10.1029/2011JD015669, 2011.

Mott, R., Faure, F., Lehning, M., Löwe, H., Hynek, B., Michlmayr, G., Prokop, A., Schöner, W., 2008. Simulation of seasonal snow cover distribution for glacierized sites (Sonnblick, Austrian Alps), Ann. Glac., 49, 155-160. Mott, R., Lehning, M., 2010. Meteorological modelling of very high resolution wind fields and snow deposition for mountains, J. Hydromet., DOI:10.1175/2010JHM1216.1.

Mott, R., Egli, L., Grünewald, T., Dawes, N., Manes. C., Bavay, M., Lehning M., 2011: Micrometeorological processes driving ablation in an alpine catchment, TC, 5, 1083-

1098, doi: doi:10.5194/tc-5-1083-2011.

Raderschall, N., Lehning, M., Schär, C., 2008. Fine scale modelling of the boundary layer wind field over steep topography, Wat. Resour. Res., 44, W09425, doi:10.1029/2007WR006544.

Stigter EE, Litt M, Steiner JF, Bonekamp PNJ, Shea JM, Bierkens MFP and Immerzeel WW (2018) The Importance of Snow Sublimation on a Himalayan Glacier. Front. Earth Sci. 6:108. doi: 10.3389/feart.2018.00108

---

## Editor Comment (EC1) · T. Mölg (Editor) · 10 Nov 2018

Dear authors,

In my access review I mentioned that there is nothing wrong with the general approach and scope of the study, yet that I see some potential problems with the WRF setup/output. Both reviewers come to a similar conclusion, and provide more details on the model setup aspect.

While I would like to note that Michael Lehning's comment on model resolution must be

viewed in light of the specific goals of a study and cannot be generalized easily (e.g., simulating many years at <100 m is not feasible in typical instances), both reviewers agree in the fact that there are major shortcomings in the setup of WRF.

Hence, my main concern is that the modeling setup was not state-of-the-art, and thus some deficiencies in the results could have been averted. Aside from spatial resolution, the specification of the land cover, the treatment of the lower boundary (SST), and the calculation of diffusion were mentioned by the reviewers. The problem with the setup makes the conclusions on WRF's downscaling abilities unreliable.

To sum up, I think it is necessary that you re-do all simulations and experiment with the WRF setup a bit more. Since this requires time and a tight deadline for revision would be counterproductive, my decision is to not consider the paper in its present version further. I do hope, however, that the reviews gave you valid input to work on an improved version of the study. In this sense, I would also like to thank the two referees for their time and thoughtful comments.

Thomas Mölg, Handling Editor & Co-Editor-In-Chief TC

---

## Author Comment (AC1) · 30 Nov 2018

**We thank the reviewer for the comments. We address them below point-by-point (our responses are in bold print).**

1. There are two key issues with the atmospheric simulations:

a. The authors have not justified their choice of model physics. The configuration

does not match any of the references on Page 6 (P6), Line 9 (L9), but in any case,

those studies focused on different mountainous and climatic regions. As the authors

mention, the choice of physics has a large impact on WRF results and should be

optimized based on a subset of the observational data or justified in some way.

**Our choice of physics schemes was informed by the previous studies on WRF that focused on SEB and glacier mass balance (Aas et al., 2015; 2016; Claremar et al 2012; Collier et al., 2013; 2015; 2018; Mölg and Kaser, 2011; Mölg et al. 2012a; 2012b). However, as the reviewer correctly noted, these studies are from different mountain regions than ours. To our knowledge, there has been no WRF application on the mountain regions where our sites lie other than Liu et al. (2016) and Wrzesien et al. (2018), which focused mainly on precipitation and SWE over a larger domain and did not analyze the WRF performance over glaciers. We are aware that WRF performance can differ due to the choice of physics schemes, as has been indicated by a large number of studies, in particular regarding the schemes used for PBL and cumulus convection (see below). We agree that more justification is needed for the schemes used in our study and we will address this issue in the revised manuscript.**

**To respond more specifically to this comment, we provide the justification (and plans for the revised manuscript) for each choice of the model physics used in our study:**

**a) Radiation: we chose RRTMG scheme as it has been the most commonly used scheme in the recent glacier studies with WRF (Aas, 2015; 2016; Collier et al 2018). We could have used CAM as in Collier et al. 2013; 2015 and Mölg and Kaser (2011), but we preferred to follow more recent studies. In our revised manuscript we plan to perform a sensitivity test that will compare RRTMG to CAM.**

**b) Planetary Boundary Layer and Surface Layer: our choice was Mellor–Yamada–Nakanishi–Niino (MLNN) level 3 (as a local closure scheme) as the scheme is shown to perform well across different regions (e.g Coniglio et al 2013; Gbode et al. 2018; Milovac et al. 2016; Shin and Dudhia, 2015). We chose MYNN2.5 surface layer because it is shown to be more compatible with the MYNN PBL scheme (Banks et al., 2016). Previous studies on glaciers mainly used non-local closure PBL schemes (e.g. YSU scheme; Aas et al., 2016; Collier et al., 2013; Mölg and Kaser, 2011), and Monin-Obukhov (MM5 revised) surface scheme. In our revised manuscript, we plan to perform sensitivity tests comparing the MMYN3+MMYN2.5 with the YSU+MM5 revised schemes.**

**c) Land surface: NOAH and NOAH-MP is the most commonly used for glacier studies (Aas et al., 1206; Collier et al 2015; 2018). Milovac et al (2016) compared the performance of NOAH-MP with NOAH and found some substantial difference in WRF performance over a domain in Germany. We used NOAH-MP, but will probably perform a sensitive test comparing this scheme with NOAH.**

**d) Cumulus: We chose Grell 3D, while an equally legit choice would be Kain-Frisch (as used in Collier et al. 2013; 2015; 2018). These two schemes have their pluses and minuses depending on what domain one is looking at, so it is a difficult choice to make. To address this more systematically, we plan to perform a**

**sensitivity test with the two schemes. There is, however, a WRF study for Korean Peninsula that suggests Grell plus Thompson microphysics scheme worked best over mountainous terrain, whereas Kain-Frisch plus Thompson microphysics worked best over the plains (Jung and Lin, 2016).**

**e) Microphysics: as noted above, our choice of Thompson scheme is compatible with Grell 3D cumulus scheme (Jung and Lin, 2016).**

b. The land surface in the closest grid cell to the observations in the ablation zones

is not classified as land ice. This inconsistency is easy to fix manually in the geo_em

files before running the WRF pre-processing programs. Manual correction in the finest

resolution domains (WRF 2.5 and 1.0) would appear to result in only one incorrect

land-use categorization at observational points, for the southern off-glacier AWS at

Castle Creek (cf. Figure 2). Reliable conclusions about the model's ability to reproduce

local meteorological conditions and katabatic flows cannot be drawn when the bottom

boundary conditions are incorrectly specified, and as a result, there are no glacierized

grid cells neighboring the one containing the station (WRF 1.0 at Castle Creek) or

there is only a single glacierized grid cell (WRF 2.5 at Nordic Glacier).

The authors attempt to address the second issue by manually changing the landsurface type in the grid point containing the AWS at Nordic Glacier. However, this

is the smallest glacier studied and may represent an underestimate of the impact of

atmosphere-glacier feedbacks on the presented results. In addition, the horizontal

resolution of the finest domain (WRF 2.5) is not well suited for this study site, as the

authors acknowledge on P27, L15. For these reasons, I think the simulations should

be repeated with accurate bottom boundary conditions (see minor comment 2 about

SST). This change would also help to streamline and simplify the manuscript (i.e., the

authors could remove P6, L29-34; P13, L17-26; P14, L24-29; Section 3.4).

**Initially, we did not want to do this (manual) correction because we trusted the ESA 300 m input data used in WRF. ESA data has the latest glacier inventory (RGI) incorporated for this region and, when plotted on 300 m grid it has all the ice/snow grid cells correctly spaced (see Figures 2 and 3 in the manuscript). However, the same data when extrapolated to 2.5 km and 1 km (ourput from geogrid) appears to have a westward shift of one grid cell for 2.5 km case and two grid cells for 1 km case (Figures 2 and 3). We recently detected the cause of this problem in the index file for ESA data, prior to running geogrid script. Thus the final output data was incorrectly extrapolated despite using the correct ESA input data. We now corrected this issue and, as a result, all AWS sites are correctly recognized as ice/snow land category. We thank the reviewer for insisting on getting this done correctly as this prompted us to identify and fix the**

**problem.**

2. The approach to the glacier simulations underutilizes the information provided by
WRF, perhaps due to the incorrect land-surface categorization. For example, for most
of the SEB simulations presented in the paper, daily mean albedo is specified from
observations for both AWS- and WRF- forced runs (e.g., P20, L1; P21, L8) rather than
using the simulated WRF value (which should be optimized in the source code, see
minor comment 9). They use a calibrated value for fresh snow density and apply a
temperature threshold for determining the frozen fraction of precipitation, however both
of these fields are available from the microphysics scheme. This approach, in particular
the albedo treatment, makes the comparison between the two forcing datasets less
informative than it could be.

**We agree with the reviewer and, especially in the light of corrected problem with the land-surface
categorization, we think it is possible to better utilize the information from WRF model output. We are
confident that we can address all key criticism received, i.e. perform the new WRF runs and sensitivity
tests accordingly. In particular, we will output frozen fraction of precipitation, albedo, sensible and latent
heat fluxes directly from WRF to be compared with our SEB model results and simple accumulation
model.**

Minor comments

1. P6, L7: Why did the authors create a new set of domains for the simulation down to
1-km grid spacing at Castle Creek? Other options would be to nest a fourth domain
or use the WRF program ndown to force two separate D3s of 2.5-km and 1-km
grid spacing. The latter would be the most consistent in terms of lateral boundary
information, and both would be more numerically efficient.

**We had some computation problems with the 1-km run setup, so we separated the two domain setups in
order to resolve this problem. In the revised manuscript we plan to do the domain setup consistently as the
reviewer suggested.**

2. P6, L15: Why was SST kept constant and how does this impact the simulations,
some of which exceed two months in length? This time-varying field is provided by
ERA-Interim and is easily incorporated into the simulations using the wrflowinp bottom
boundary updates.

**Similarly to the above comment, we had some computation problems initially with the changing SST
boundary conditions, which prompted us to perform the runs with the SST kept constant. We are,
however, aware that this is far from ideal. In the revised manuscript we plan to use changing SST.**

3. P6, L24: Please provide the exact spin-up time and a reason for this choice.

**We will do so in the revised manuscript.**

4. P7, Table 3:

a. Are the timesteps correct? If yes, is the model solution stable and physical with a

timestep of 20 times the grid spacing at Castle Creek?

**There is a typo. Timesteps should be corrected to: 75, 15, 3 s.**

b. In complex terrain, diffusion should be computed in physical space (diff_opt set to

2) for more accurate results where coordinate surfaces are sloped.

**We thank the reviewer for this comment. We will follow this advise in the revised WRF setup.**

5. P8, Figure 2: The authors state that they updated the land-use data using ESA CCI,

however certain areas that appear to be at least 506. P9/10: The paragraph explaining

the bulk aerodynamic method could be removed, since it is well established and the

reader is referred to Fitzpatrick et al. (2017).

**As stated above we will be using corrected ESA data (corrected index file) so these results will most likely be revised. We will consider removing the paragraph explaining the bulk method.**

7. P10, L15-16: What ice/snow albedos are supported by the measurements?

**The chosen albedo values (in the SEB model) agree with those observed at our sites (see Figure 4).**

8. P16: I suggest showing the comparison for precipitation, as it plays a role later in

the modelled surface height changes.

**This will be addressed in the revised manuscript.**

9. P24, L8: The albedo for glacierized grid cells is a prognostic variable in WRF. The

default lower bound (variable ALBICE in phys/module_sf_noahmp_glacier.F) is set

unrealistically high at 0.67 and should be changed to a value consistent with bare ice

for more realistic simulations and atmospheric feedbacks.

**This will be addressed in the revised manuscript. Note that, due to corrected land-surface categorizations, out sensitivity tests will not be the same.**

10. P26, L2: Please rephrase to match what is described in the methods.

**This will be addressed in the revised manuscript.**

11. P28, L17: Further optimization may be possible. For comparison, we run a

three-domain configuration down to sub-kilometer grid spacing with dimensions

exceeding 300x300x50 and are able to complete more than 20 simulation days in one

day of wall-time on 500 processors.

**We thank the reviewer for pointing this out. This computational efficiency with 500 processors is indeed impressive. As a comparison, we operate with 30-50 processors and the given example would take us 10 days.**

Technical comments

1. I suggest changing "near-surface" to "surface" in the title, as only surface mass and

energy fluxes are considered.

2. P1, L5: for clarity, I suggest changing "nested within the ERA-Interim" to "forced at

its lateral boundaries by ERA-Interim."

3. P1, L6: change "spatial resolution" to "grid spacing."

4. Section 2.1: I suggest referencing Figures 2 and 3 where applicable to make it

easier for the reader to follow the station locations.

5. P6, L2: "advanced research version of the WRF model."

6. Throughout the paper, please change "(see text)" to refer to the relevant section.

7. P14, L28 and elsewhere: I think the phrase "(not shown)" is overused in the

manuscript. I suggest removing some of the statements if they are not important or

introducing supplementary material.

8. P27, L4-5: This sentence appears nearly word-for-word in Collier et al. (2013).

Tables Figures

Figure 1: I suggest adding shaded model topography.

Figures 2 and 3: Please label the axes or provide a scale. For Figure 3, is there a pink

triangle?

Table 3: Please provide the grid dimensions.

**All technical comments above will be addressed in the revised manuscript.**

**References:**

**Aas, K. S., T. Dunse, E. Collier, T. V. Schuler, T. K. Berntsen, J. Kohler, and B. Luks (2016): The climatic mass balance of Svalbard glaciers: a 10-year simulation with a coupled atmosphere-glacier mass balance model. Cryosphere, 10(3):1089–1104.**

**Aas, K. S. , Berntsen, T. K. , Boike, J. , Etzelmüller, B. , Kristjánsson, J. E. , Maturilli, M. , Schuler, T. V. , Stordal, F. and Westermann, S. (2015): A Comparison between Simulated and Observed Surface Energy Balance at the Svalbard Archipelago , Journal of Applied Meteorology and Climatology, 54 (5), pp. 1102-1119 . doi: 10.1175/JAMC-D-14-0080.1**

Banks R.F., Jordi Tiana-Alsina, José María Baldasano, Francesc Rocadenbosch, Alexandros Papayannis, Stavros Solomos, Chris G. Tzanis, (2016) Sensitivity of boundary-layer variables to PBL schemes in the WRF model based on surface meteorological observations, lidar, and radiosondes during the HygrA-CD campaign, Atmospheric Research, 176–177, 185-201.

Claremar, B., F. Obleitner, C. Reijmer, V. Pohjola, A. Waxegard, F. Karner, and A. Rutgersson (2012): Applying a Mesoscale Atmospheric Model to Svalbard Glaciers. Advances in Meteorology, 2012:1–22

Collier, E., T. Mölg, F. Maussion, D. Scherer, C. Mayer, and A. Bush (2013): High-resolution interactive modelling of the mountain glacier-atmospheric interface: an application over the Karakoram. The Cryosphere, 7:779–795

Collier, E., F. Maussion, L. Nicholson, T. Molg, W. Immerzeel, and A. Bush (2015): Impact of debris cover on glacier ablation and atmospheric-glacier feedbacks in the Karakoram. The Cryosphere, 9:1617–1632.

Collier, E., Mölg, T., and Sauter, T. (2018): Recent atmospheric variability at Kibo Summit, Kilimanjaro, and its relation to climate mode activity, J. Clim, 31, 3875–3891, doi:10.1175/JCLI-D-17-0551.1

Coniglio C., Michael & Correia, Jr, James & Marsh, Patrick & Kong, Fanyou (2013): Verification of Convection-Allowing WRF Model Forecasts of the Planetary Boundary Layer Using Sounding Observations. Weather and Forecasting. 28. 10.1175/WAF-D-12-00103.1.

Gbode, I.E., Dudhia, J., Ogunjobi, K.O. and Ajayi V. O. (2018) Sensitivity of different physics schemes in the WRF model during a West African monsoon regime, Theor Appl Climatol., https://doi.org/10.1007/s00704-018-2538-x

Jung, Yong & Lin, Yuh-Lang. (2016): Assessment of a Regional-Scale Weather Model for Hydrological Applications in South Korea. Environment and Natural Resources Research. 6. 28. 10.5539/enrr.v6n2p28.

Liu, C., Ikeda, K., Rasmussen, R., Barlage, M., Newman, A. J., Prein, A. F., et al. (2016): Continental-scale convection-permitting modeling of the current and future climate of North America. Climate Dynamics, 49(1-2), 71–95. https://doi.org/10.1007/s00382-016-3327-9

Milovac, J., K. Warrach-Sagi, A. Behrendt, F. Späth, J. Ingwersen, and V. Wulfmeyer (2016), Investigation of PBL schemes combining the WRF model simulations with scanning water vapor differential absorption lidar measurements,J. Geophys. Res. Atmos., 121, 624–649, doi:10.1002/2015JD023927.

Mölg, T. and Kaser, G. (2011): A new approach to resolving climate-cryosphere relations: Downscaling slimate dynamics to glacier-scale mass and energy balance without statistical scale linking, Journal ofGeophysical research, 116, 779–795, https://doi.org/10.1029/2011JD015669

Mölg, T., Großhauser, M., Hemp, A., Hofer, M., and Marzeion, B. (2012a): Limited forcing of glacier loss through land-cover change on Kilimanjaro, Nature Climate Change, 2(4), 254–258, https://doi.org/10.1038/NCLIMATE1390

Mölg, T., Maussion, F., Yang, W., and D., S. (2012b): The footprint of Asian monsoon dynamics in the mass and energy balance of a Tibetan glacier, The Cryosphere, 6, 1445–1461, https://doi.org/10.5194/tc-6-1445-2012

Shin, H. H., and J. Dudhia (2016): Evaluation of PBL parameterizations in WRF at subkilometer grid spacings: Turbulence statistics in the dry convective boundary layer. Monthly Weather Review, 144, 1161-1177, doi:10.1175/MWR-D-15-0208.1.

Wrzesien, M. L., Durand, M. T., Pavelsky, T. M., Kapnick, S. B., Zhang, Y., Guo, J., & Shum, C. K. (2018). A new estimate of North American mountain snow accumulation from regional climate model simulations. Geophysical Research Letters, 45, 1423–1432. https:// doi.org/10.1002/2017GL076664

---

## Author Comment (AC2) · 30 Nov 2018

**We thank the reviewer for his comments. We address them below point-by-point (our responses are in bold font).**

The set-up of WRF for such a downscaling analysis, the collection of related field data and the analysis of model in comparison to the measurements is a lot of work and a great effort. This is fully acknowledged but at the same time, the set-up of the study is such that the negative results (no improvement through WRF) are no surprise. The study is heavily outdated in a time when weather models run at 2 km resolution operationally (e.g. Cosmo in Switzerland) and WRF downscaling work is done at the resolution of tens of meters (Gerber et al., 2018) and not kilometers as in the submitted paper. A study that has just been published and uses a 30 m resolution to generate meteorological input for glacier energy and mass balance in the Himalayas is Stigter et al. (2018) and is showing the state of the art in the field and in fact documents that WRF is able to successfully generate useful local weather input.

From our own early work (Lehning et al. 2008; Raderschall et al., ), we know that it needs a very high resolution of below 50 m to approximate wind fields in complex terrain. And if you have correct high-resolution wind fields, you can describe boundary layer processes from snow deposition (Mott and Lehning, 2010) to snow ablation (Mott et al., 2011). I would go as far as to say that the dynamical downscaling of first flow and then full weather has been initiated with these ARPS based studies and the methodology has immediately also been applied to glaciers (Mott et al., 2008; Dadic et al., 2010) including the Mölg and Kaser study (Mölg and Kaser, 2011) mentioned in the paper. Therefore, if you do a downscaling study that will not reproduce the local wind fields correctly (in this case katabatic flows), you cannot expect to see much improvement over a re-analysis on local snow mass balance estimates. Now, if this was the first attempt and all that is currently possible, then this could still be interesting. But if much older and current studies (such as the ones mentioned above) already have pushed the limits way beyond the setup of this study and state of the art downscaling in much higher resolution shows that you can even simulate individual eddies (Gerber et al., 2017) then the results as presented in the paper do not add to our scientific body of knowledge.

Let me please emphasize that I am not trying to reject the paper because our own work

and other work as discussed above did not get cited. This may be seen as an omission

but is not even necessary or could be fixed easily. The main point is really that the

results do not allow to gain new insight and give a wrong impression on the usefulness

of dynamical downscaling because the study had the wrong/outdated design despite

all the good work that has been done in the execution. I would also fully be in favor

of publishing negative results but not if the negative results are the consequence of an

inadequate set-up such as in this study. This is unfortunate, as the paper is really well

written and nicely illustrated.

**After carefully reading the reviewer's comments we realized that our objectives should have been more clearly stated. Specifically, our goal was not to use the highest possible resolution in WRF to simulate surface energy balance (SEB) at a given point on a glacier surface and to resolve turbulent eddies. Instead, our ultimate goal, as will be stated more clearly in the revised manuscript is to develop a regional glaciation modelling approach that would incorporate SEB modelling forced by coupled dynamical and statistical downscaling. The first step toward this ultimate goal, addressed in this study, is to evaluate the performance of a SEB model forced with dynamically downscaled fields at three glaciers in BC that were the subject of multi-year observations of all SEB components. With the use of WRF, we downscale meteorological variables and energy fluxes at spatial resolution that can be computationally attainable for large spatial domains (e.g. all mountain ranges in BC an Alberta) and relatively long periods (e.g. a decade). As an example of this attainability we refer to Liu et al (2016), a study that produced high-resolution downscaled climate fields at 4-km grid spacing over much of North America for 13-year period in present and future climate. In our study we downscale ERA-Interim to 2.5 km grid at all three glaciers, and further to 1 km at one glacier, for >200 days in total (four ablation seasons).**

**The reviewer has rejected our manuscript on the basis that our study, in particular the WRF setup and resolution, is 'heavily outdated', arguing that some recent studies (focused on snow processes) run WRF at 50 or 30 m resolution (citations of the reviewer's own work provided). While the cited studies represent indeed an impressive piece of work, they all deal with much smaller spatial scales and time periods (2 to 3 days of WRF simulations in total) than those we targeted and of relevance for our ultimate goal (regional glaciation modelling). We note that the recent downscaling work that focused on SEB and glacier mass balance (Aas et al., 2015; 2016; Claremar et al 2012; Collier et al., 2013; 2015; 2018; Mölg and Kaser, 2011; Mölg et al. 2012A; 2012b) and the ongoing work on climate downscaling over large complex terrain (e.g. Jung and Lin, 2016; Wrzesien et al., 2017; 2018) have been performed on the same scale (one kilometer or few kilometres) as in our study, so there is nothing outdated or wrong with our grid-spacing setup given the scale of the forcing.**

**We agree with the reviewer #1 and the editor that WRF needs to be re-run because there were some recognized inconsistencies in the model setup. We are confident that we can address all key criticism we received, i.e. perform the new WRF runs and sensitivity tests accordingly. In particular, we will:**
**1. justify our choice of model physics**
**2. run WRF with corrected land cover; all runs will have changing SST as boundary conditions; diffusion will be calculated in physical space**
**3. run WRF with consistent 2.5 km grid inner domain for all sites; and a further 850 m grid inner domain**

**for one site (Castle glacier)**
**4. output frozen fraction of precipitation, albedo, sensible and latent heat fluxes directly from WRF to be compared with our SEB model results and simple accumulation model**

In addition to the general comments, I have also one additional major set-up problem,

which is the arbitrary switch-off of the cumulus convection scheme, while it is quite

clear that convection will be insufficiently resolved at a 2.5 km grid resolution. Again, it

then no surprise that precipitation simulations have a large error.

**We disagree that this qualifies as a problem. The cumulus parametrization can be turned off below 4 km model resolution (see Weisman et al. 1997). The rule of thumb (see WRF current documentation and guidelines from NCAR) is to switch it off below 5 km resolution, but certainly at 2.5 km and below the parametrization is not needed as the model is eddy 'permitting' (not resolved, but permitted; meaning that the eddies are there but not greatly resolved). Please note the previous studies on WRF application on glaciers also switched off the cumulus convection scheme for the inner-most domain with resolution below 5 km (Aas et al., 2015; 2016; Claremar et al 2012; Collier et al., 2013; 2015; 2018; Mölg and Kaser, 2011; Mölg et al. 2012A; 2012b)**

One final major point, which can either be a typo or a serious misconception is the

statement on p. 27 l. 10 when the authors talk about adiabatic cooling in the katabatic

wind on the glacier. Of course, descending air masses warm by an adiabatic process.

**This is indeed a typo; we thank the reviewer for spotting it. We meant 'advective cooling' as stated on page 22, line 7. The working assumption here is that the cold air from glacier accumulation area drains non-adiabatically downslope. The air is expected to warm up adiabatically as it descents downslope, but because the shallow jet of air exchanges heat in contact with the glacier surface (which cannot exceed 0ºC), the air warms up less than it would if the process was adiabatic -> this results in advective cooling at the lower station (in the ablation area).**

Some detailed comments:

p. 5 l. 6: Roughness lengths should be consistent with model resolution as sub-grid

topography needs to be represented by the roughness

**Please note that we did not alter the default roughness lengths in the WRF model, i.e. in the Noah-MP land surface model. Thus there is no inconsistency with WRF model resolution in our model runs. As explained in the text, the roughness lengths (calculated from the eddy-covariance measurement at each site) are only used in our SEB model, which is run off-grid with WRF output data.**

p. 5 l.8: Why did you not use a precipitation lapse rate?

**The point was to make a direct comparison of WRF output with observations that are not altered or corrected in any way (bias corrections or application of assumed lapse rates).**

p. 9 Eq. 2: I don't know why there is p/p_0 in this equation. This does not come from

the original derivation of the bulk formula (see e.g. textbooks by Brutsaert or Stull) and

the influence of air pressure is already there via the air density.

**This is a typo; we thank the reviewer for spotting it. The air density (ρ_a) at each site is derived as the air density at standard sea-level pressure (ρ_0 = 1.29 kg/m3 at 0ºC) multiplied by the ratio between the air pressure at each site (p) and the standard sea-level pressure (p_0 ; 1013 hPa). So it should be ρ_0 instead of ρ_a in these equations.**

p. 19 l.11: There is some physical argument why roughness for momentum could be

different from roughness of scalars (but empirical evidence is missing, see e.g. Schlögl

et al., 2017). However, assuming two different roughness lengths for moisture and

temperature has no theoretical justification

**We agree that there is no theoretical justification. However, we use the roughness lengths (seasonal mean) as assessed from our eddy-covariance data from each site. Thus we use the empirical values for all three roughness lengths and this empirical values show difference between roughness for humidity and temperature.**

p. 13 l.3: You could diagnose local stability (at least over a melting surface – not

sure you had a surface temperature measurement) and then use an adequate stability

correction.

**Please note that we do diagnose local stability (z/L) which is used in the bulk method with the stability corrections to derive turbulent heat fluxes (page 10, line 6-10). It is true that we could use the same parameter (z/L) to diagnose whether a log-linear or log profile of wind applies, and then perform the wind corrections accordingly. However, due to the poor resemblance between diagnosed and observed (eddy-covariance derived) z/L (see Fitzpatrick et al, 2017; Radic et al., 2017) we chose not to introduce any additional uncertainty in the wind profiles (either observed or WRF-derived) by implementing this potential wind corrections.**

**References:**

**Aas, K. S., T. Dunse, E. Collier, T. V. Schuler, T. K. Berntsen, J. Kohler, and B. Luks (2016): The climatic mass balance of Svalbard glaciers: a 10-year simulation with a coupled atmosphere-glacier mass balance model. Cryosphere, 10(3):1089–1104.**

**Aas, K. S. , Berntsen, T. K. , Boike, J. , Etzelmüller, B. , Kristjánsson, J. E. , Maturilli, M. , Schuler, T. V. , Stordal, F. and Westermann, S. (2015): A Comparison between Simulated and Observed Surface Energy Balance at the Svalbard Archipelago , Journal of Applied Meteorology and Climatology, 54 (5), pp. 1102-1119 . doi: 10.1175/JAMC-D-14-0080.1**

**Claremar, B., F. Obleitner, C. Reijmer, V. Pohjola, A. Waxegard, F. Karner, and A. Rutgersson (2012): Applying a Mesoscale Atmospheric Model to Svalbard Glaciers. Advances in Meteorology, 2012:1–22**

Clarke, G. K., Jarosch, A. H., Anslow, F. S., Radic, V., and Menounos, B. (2015): Projected deglaciation of western Canada in the twenty-first century, Nature Geoscience, 8, 372–377, https://doi.org/10.1038/NGEO2407.

Collier, E., T. Mölg, F. Maussion, D. Scherer, C. Mayer, and A. Bush (2013): High-resolution interactive modelling of the mountain glacier-atmospheric interface: an application over the Karakoram. The Cryosphere, 7:779–795

Collier, E., F. Maussion, L. Nicholson, T. Molg, W. Immerzeel, and A. Bush (2015): Impact of debris cover on glacier ablation and atmospheric-glacier feedbacks in the Karakoram. The Cryosphere, 9:1617–1632.

Collier, E., Mölg, T., and Sauter, T. (2018): Recent atmospheric variability at Kibo Summit, Kilimanjaro, and its relation to climate mode activity, J. Clim, 31, 3875–3891, doi:10.1175/JCLI-D-17-0551.1

Fitzpatrick, N., Radic, V., and Menounos, B. (2017): Surface Energy Balance Closure and Turbulent Flux Parameterization on a Mid-Latitude Mountain Glacier Purcell Mountains, Canada, Frontiers in Earth Science, 5, https://doi.org/10.3389/feart.2017.00067.

Jung, Yong & Lin, Yuh-Lang. (2016): Assessment of a Regional-Scale Weather Model for Hydrological Applications in South Korea. Environment and Natural Resources Research. 6. 28. 10.5539/enrr.v6n2p28.

Liu, C., Ikeda, K., Rasmussen, R., Barlage, M., Newman, A. J., Prein, A. F., et al. (2016): Continental-scale convection-permitting modeling of the current and future climate of North America. Climate Dynamics, 49(1-2), 71–95. https://doi.org/10.1007/s00382-016-3327-9

Mölg, T. and Kaser, G. (2011): A new approach to resolving climate-cryosphere relations: Downscaling slimate dynamics to glacier-scale mass and energy balance without statistical scale linking, Journal ofGeophysical research, 116, 779–795, https://doi.org/10.1029/2011JD015669

Mölg, T., Großhauser, M., Hemp, A., Hofer, M., and Marzeion, B. (2012a): Limited forcing of glacier loss through land-cover change on Kilimanjaro, Nature Climate Change, 2(4), 254–258, https://doi.org/10.1038/NCLIMATE1390

Mölg, T., Maussion, F., Yang, W., and D., S. (2012b): The footprint of Asian monsoon dynamics in the mass and energy balance of a Tibetan glacier, The Cryosphere, 6, 1445–1461, https://doi.org/10.5194/tc-6-1445-2012,

Radic, V., Menounos, B., Shea, J., Fitzpatrick, N., Tessema, M. A., and Dery, S. J. (2017): Evaluation of different methods to model near-surface turbulent fluxes for a mountain glacier in the Cariboo Mountains, BC, Canada, Cryosphere, 11, 2897–2918, https://doi.org/10.5194/tc-11- 2897-2017

Weisman, M.L., W.C. Skamarock, and J.B. Klemp (1997): The Resolution Dependence of Explicitly Modeled Convective Systems. Mon. Wea. Rev., 125, 527–548, https://doi.org/10.1175/1520-0493(1997)125

Wrzesien, M. L., Durand, M. T., Pavelsky, T. M., Kapnick, S. B., Zhang, Y., Guo, J., & Shum, C. K. (2018). A new estimate of North American mountain snow accumulation from regional climate model simulations. Geophysical Research Letters, 45, 1423–1432. https:// doi.org/10.1002/2017GL076664

Wrzesien, M. L., Durand, M. T., Pavelsky, T. M., Howat, I. M., Margulis, S. A., & Huning, L. S. (2017). Comparison of methods to estimate snow water equivalent at the mountain range scale: A case study of the California Sierra Nevada. Journal of Hydrometeorology, 18(4), 1101–1119. https://doi.org/10.1175/JHM-D-16-0246.1